# Revisiting Feature Prediction for Learning Visual Representations from Video

## Abstract

This paper explores feature prediction as a stand-alone objective for unsupervised learning from video and introduces V-JEPA, a collection of vision models trained solely using a feature prediction objective, without the use of pretrained image encoders, text, negative examples, reconstruction, or other sources of supervision. The models are trained on 2 million videos collected from public datasets and are evaluated on downstream image and video tasks. Our results show that learning by predicting video features leads to versatile visual representations that perform well on both motion and appearance-based tasks, without adaption of the model's parameters; e.g., using a frozen backbone. Our largest model, a ViT-H/16 trained only on videos, obtains 81.9% on Kinetics-400, 72.2% on Something-Something-v2, and 77.9% on ImageNet1K.

## 1 Introduction

Humans possess the remarkable ability to map low-level signals originating from the retina into a semantic spatio-temporal understanding of the world; synthesizing notions such as objects and global motion (Spelke et al., 1995). A long-standing goal of the machine learning community is to identify the principles or objectives that may guide such unsupervised learning in humans (Field, 1994; Berkes & Wiskott, 2005; Hinton, 1989). One related hypothesis is based on the *predictive feature principle* (Rao & Ballard, 1999), which posits that representations of temporally adjacent sensory stimuli should be predictive of each other.

In this work, we revisit feature prediction as a stand-alone objective for unsupervised learning of visual representations from video. Numerous advances in the field — such as the standard use of transformer architectures in vision (Dosovitskiy et al., 2020), the maturing of masked autoencoding frameworks (Xie et al., 2021; Bao et al., 2021; He et al., 2021), query-based feature pooling (Chen et al., 2022), joint-embedding predictive architectures (JEPA) (LeCun, 2022; Assran et al., 2023; Baevski et al., 2022b), and larger datasets — form a unique arsenal of tools, which we integrate in a modern and conceptually simple method, the *video joint-embedding predictive architecture* or V-JEPA, which is based solely on feature prediction, without using pretrained image encoders, text, negative examples, human annotations, or pixel-level reconstruction.

We seek to answer the simple question:

> *How effective is feature prediction as a stand-alone objective for unsupervised learning from video with modern tools?*

To that end, we pretrain a family of V-JEPA models on a dataset of 2 million videos collected from publicly available datasets by combining a masked modeling prediction task with a joint-embedding predictive architecture (see Figure 2). We measure performance on several downstream image and video tasks, using both frozen evaluation and end-to-end fine-tuning. Our findings suggest that feature prediction can indeed serve as an effective stand-alone objective for unsupervised learning from video, while using significantly shorter training schedules than pixel prediction methods. Specifically:

- Feature prediction leads to versatile visual representations that perform well across downstream image and video tasks without adaption of the model's weights; i.e., using a frozen backbone. V-JEPA achieves

Figure 1: V-JEPA models pretrained on video learn versatile visual representations. It performs well on motion-based tasks (Something-Something-v2) and appearance-based tasks (Kinetics 400) without adaptation of the model's parameters, i.e., using the same frozen backbone for both tasks.

the best performance among methods we consider (+6% accuracy) on the SomethingSomething-v2 task, which requires fine-grained temporal understanding. V-JEPA is also competitive on tasks like Kinetics400, where appearance-based features are sufficient and hence state-of-the-art image models such as DINOv2 excel (Figure 1 and Table 6).

- Models trained with feature prediction are superior to pixel prediction approaches under a frozen evaluation protocol (attentive probing) and are competitive with pixel prediction under full fine-tuning, while using significantly shorter training schedules (Tables 5 and 6).

- Models trained with feature prediction are more label-efficient than pixel prediction approaches. Decreasing the available number of labeled examples results in an increase in the performance gap between V-JEPA and pixel-reconstruction models (Table 8).

## 2 Related Works

**Slow Features.**  One way to encourage temporally adjacent representations to be predictive of each other is to ensure that they vary slowly over time. Early works targeting predictive features encouraged representations of individual video frames to be locally temporally invariant, while preventing representation collapse by using spectral methods, as in SFA (Wiskott & Sejnowski, 2002), SSA (Kayser et al., 2001), and Simulated Fixations (Zou et al., 2012). More recently, Goroshin et al. (2015); Wang et al. (2010) train a siamese convolutional network to map the representations of two subsequent frames to the same point, while encouraging distant frames to have diverse representations via a pair-wise margin loss and a triplet loss, respectively. Other works (Oord et al., 2018; Surís et al., 2021; Feichtenhofer et al., 2021) implement temporal invariance using noise-contrastive estimation (Gutmann & Hyvärinen, 2012). Our exploration in this paper goes beyond temporal invariance and explores feature prediction using masked modeling.

**Predictive Features.**  Going beyond local invariance, a family of works trains a predictor network to map the representation of a frame or clip at one time-step to a distinct representation at another time-step. Srivastava et al. (2015); Vondrick et al. (2016); Wang et al. (2023b) train such a video feature predictor network on top of a frozen pretrained image or video encoder. Unfreezing the target feature extractor, several methods train the video encoder and the predictor network simultaneously, while preventing collapse by using a supervised action forecasting loss (Girdhar & Grauman, 2021), or by using the representations of distant

clips as negative samples in a contrastive loss (Han et al., 2019; 2020; Tan et al., 2023), often focusing on small convolutional encoders (Han et al., 2019; 2020). The idea of learning a representation by predicting missing information in feature space is also core to the joint-embedding predictive architecture (JEPA) (LeCun, 2022), which combines a siamese encoder with a predictor network. JEPAs have been successfully instantiated in several modalities, such as with audio data (Baevski et al., 2022b) and image data (Zhou et al., 2021; Oquab et al., 2023; Assran et al., 2023). In this work, we extend this paradigm to video data by leveraging recent advances in self-supervised learning.

**Advances in Self-Supervised Learning.** The use of vision transformers (Dosovitskiy et al., 2020; Li et al., 2022) has become standard practice in self-supervised learning with joint-embedding architectures (Chen et al., 2021; Caron et al., 2021; Oquab et al., 2023; Zhou et al., 2021; Assran et al., 2022), and unlocked masked image modeling in pixel space by parameterizing the pixel decoder as a transformer with learnable mask tokens (Dosovitskiy et al., 2020; Xie et al., 2021; He et al., 2021; Bao et al., 2021), demonstrating a step-change in the representation quality of autoencoding methods (Vincent et al., 2010). This line of generative methods was subsequently extended to video data using spatio-temporal masking (Tong et al., 2022; Feichtenhofer et al., 2022; Wang et al., 2023a; Kalluri et al., 2023; Gupta et al., 2023). It was also recently shown that the representations of masked image autoencoders could be significantly improved by using learnable pooling mechanisms based on cross-attention (Chen et al., 2022). Finally, through careful selection of design choices, the non-contrastive collapse prevention strategy in BYOL (Grill et al., 2020) was recently made to work with image feature prediction methods (Baevski et al., 2022b; Assran et al., 2023), which demonstrated the ability to learn representations that can be leveraged for various downstream tasks without relying on invariance to hand-crafted image transformations.

**Feature Prediction versus Pixel Reconstruction.** Approaches that predict in pixel space must dedicate significant model capacity and compute to capture all the low-level detail in the visual input. By contrast, approaches that predict in latent space have the flexibility to eliminate irrelevant or unpredictable pixel-level details from the target representation (Vondrick et al., 2016). Predicting in representation space has been shown to lead to versatile representations that perform well across many downstream tasks through linear probing or low-shot adaptation (Assran et al., 2023; Oquab et al., 2023; Assran et al., 2022), while demonstrating an efficiency gain during pretraining compared to pixel level reconstruction (Assran et al., 2023; Baevski et al., 2022b;a). The works of Baevski et al. (2022a;b) additionally show that predicting in representation space results in competitive end-to-end fine-tuning performance in the image, audio and text domains. In this work, we extend these findings to the video modality.

## 3 Methodology: Video-JEPA

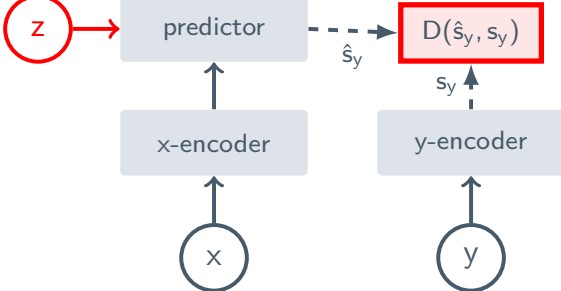

Figure 2: Joint-Embedding Predictive Architectures are trained to predict the representation of an input $y$ from the representation of another input $x$. The additional variable $z$ provides the predictor with information about the transformation that computes $y$ from $x$.

Our goal is to explore the effectiveness of feature prediction as a stand-alone objective for learning visual representations from video. To that end, we use a joint-embedding predictive architecture (JEPA) (LeCun, 2022); see Figure 2. The main idea behind a JEPA is to learn by predicting the representation of an input $y$

from the representation of another input $x$. The basic architecture is made up of an encoder, $E_\theta(\cdot)$, which computes the representation of the inputs, and a predictor, $P_\phi(\cdot)$, which predicts the representation of $y$ from the representation of $x$, conditioned on a variable $z$ indicating the transformation (or corruption) between $x$ and $y$. Conditioning on $z$ enables the generation of distinct predictions for various transformations of $x$.

### 3.1 Training Objective

We train our visual encoder $E_\theta(\cdot)$ to satisfy the constraint that representations computed from one part of the video, $y$, should be predictable from representations computed from another part of the video, $x$. The predictor network $P_\phi(\cdot)$, which maps the representation of $x$ to the representation of $y$, is trained simultaneously with the encoder, and is provided specification of the spatio-temporal positions of $y$ through the conditioning variable $z \leftarrow \Delta_y$.

Naively implementing the objective using the regression

$$\text{minimize}_{\theta,\phi} \quad \|P_\phi(E_\theta(x), \Delta_y) - E_\theta(y)\|_1,$$

would admit a trivial solution, where the encoder outputs a constant representation, regardless of its input. In practice, we use the following modified objective to prevent representation collapse,

$$\text{minimize}_{\theta,\phi} \quad \|P_\phi(E_\theta(x), \Delta_y) - \text{sg}(\overline{E}_\theta(y))\|_1, \tag{1}$$

where $\text{sg}(\cdot)$ denotes a stop-gradient operation, which does not backpropagate through its argument, and $\overline{E}_\theta(\cdot)$ is an exponential moving average of the network $E_\theta(\cdot)$. The use of an exponential-moving average feature extractor along with a stop-gradient and a predictor has been used as a collapse prevention strategy for image pretraining (Grill et al., 2020), and studied empirically (Xie et al., 2021) and theoretically (Tian et al., 2021). In fact, the objective in equation equation 1 is similar to the loss of Assran et al. (2023) used for image pretraining, but we modify it to use an $\ell_1$ regression, which we found to be more stable.

**Theoretical motivation.** A theoretical motivation for the effectiveness of this collapse prevention strategy was proposed in Grill et al. (2020) for the BYOL method. We provide a simple adaptation of their analysis for our $\ell_1$ loss. For ease of exposition, we will disregard the effect of the conditioning variable $z$ and consider one dimensional representations. Denote the representation $\overline{E}_\theta(y)$ by a random variable $Y$. The optimal predictor under equation equation 1 is thus given by the following functional expression,

$$P^\star(E_\theta(x)) = \text{argmin}_P \|P(E_\theta(x)) - Y\|_1$$
$$= \text{median}(Y|E_\theta(x)).$$

Substituting this expression for the optimal predictor into the loss function and evaluating the expected gradient of the encoder gives

$$\nabla_\theta \mathbb{E}\|P^\star(E_\theta(x)) - Y\|_1 = \nabla_\theta \text{MAD}(Y|E_\theta(x)),$$

where $\text{MAD}(\cdot \,|E_\theta(x))$ is the median absolute deviation of a random variable conditioned on $E_\theta(x)$. Thus, in the case where the predictor is optimal, the encoder must learn to capture as much information about the video as possible to minimize the deviation of the target. The hypothesis is that incorporating an exponential moving average to compute the representation of $y$ ensures that the predictor evolves faster than the encoder and remains close to optimal, thereby preventing collapse.

### 3.2 Prediction Task: Predicting $y$ from $x$

The feature prediction task is based on a masked modeling formulation (He et al., 2021; Tong et al., 2022); i.e., regions $x$ and $y$ from the video are sampled using masking. To sample $y$ from a video, we sample several (possibly overlapping) spatially continuous blocks with various aspect ratios and repeat the spatial blocks across the entire temporal dimension of the video; $x$ is taken to be the complement. Masking a large continuous block that covers the full temporal dimension limits information leakage due to the spatial and temporal redundancy of videos, and results in a harder prediction task (Tong et al., 2022).

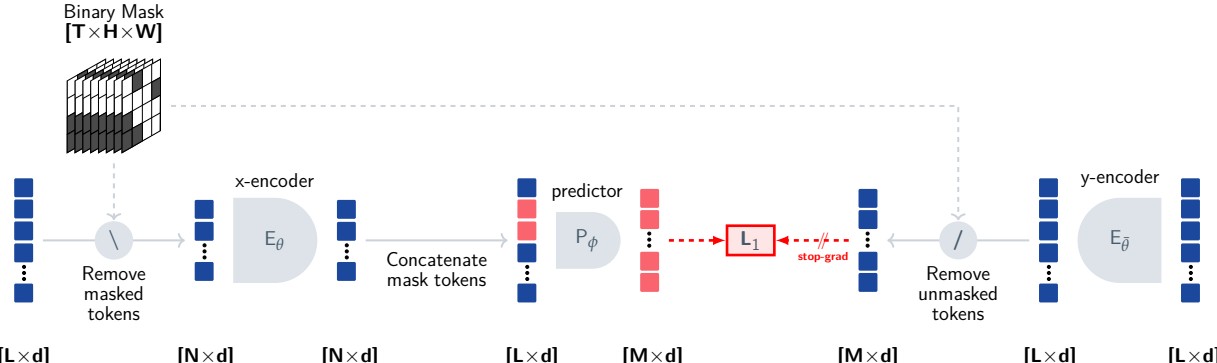

Figure 3: *V-JEPA.* Training operates on a video clip of $T$ frames with spatial resolution $H \times W$, flattened into a sequence of $L$ tokens. (Left to right): We first obtain the input of the $x$-encoder by dropping tokens from the video clip. The $x$-encoder then processes the masked video sequence, and outputs an embedding vector for each input token. Next, the outputs of the $x$-encoder are concatenated with a set of learnable mask tokens containing positional embeddings of the masked spatio-temporal patches. The predictor network processes the combined token sequence, and outputs an embedding vector for each mask token. The outputs of the predictor are then regressed to the prediction targets using an $L_1$ loss. The prediction targets correspond to the output of the $y$-encoder.

We leverage two types of masks: short-range masks, where we take the union of 8 randomly sampled target blocks covering 15% of each frame, and long-range masks, where we take the union of 2 randomly sampled target blocks covering 70% of each frame. In both cases, the aspect ratio for all sampled blocks is randomly chosen in the range $(0.75, 1.5)$. Given that both short-range and long-range masks are produced by sampling many blocks and taking their union, the result is an average masking ratio of $\sim 90\%$. We refer to our masking strategy as multi-block and compare it to other possible masking strategies in Section 4. Refer to Appendix B for more details on the multiblock implementation.

### 3.3 Network Parameterization

We use a Vision Transformer (ViT) (Dosovitskiy et al., 2020; Arnab et al., 2021) as our video backbone. To process a video with a transformer network, we split the video clip into a 3D grid of $L$ spatio-temporal patches, where a patch consists of a $16 \times 16$ pixel block spanning 2 consecutive frames; we refer to these spatio-temporal patches as tokens. This sequence of tokens is then directly processed by the stack of transformer blocks. Inputs $x$ and $y$ correspond to masked regions of a video, we apply the video masks by simply dropping a subset of the tokens. We apply masking at the input of the $x$-encoder, and at the output of the $y$-encoder to construct contextualized targets (Baevski et al., 2022b). The encoder is parameterized using standard ViT networks, while the predictor is a narrow transformer implemented using 12 blocks with an embedding dimension of 384. Taking inspiration from masked autoencoders (He et al., 2021), our predictor takes as input the sequence of embeddings produced by the $x$-encoder as well as a sequence of learnable mask tokens with positional embeddings indicating the spatio-temporal positions of the $y$ tokens. The sequence of mask tokens implement the conditioning variable $z$ (see Figure 2) which allows for outputting different representations for different spatio-temporal positions. The output of the predictor is an embedding vector for each mask token; see Figure 3 and refer to Appendix B for more details.

### 3.4 Pretraining Data and Evaluation Setup

**Pretraining.** We combine several public datasets to construct an unsupervised video pretraining dataset, which we refer to as VideoMix2M. Specifically, we combine the videos from HowTo100M (HT) (Miech et al., 2019), Kinetics-400/600/700 (K710) (Kay et al., 2017), and Something-Something-v2 (SSv2) (Goyal et al., 2017), and remove any overlap with the validation sets of Kinetics-400/600/700 and Something-Something-v2, resulting in approximately 2 million videos. We train a ViT-L/16, a ViT-H/16, and a ViT-H/16$_{384}$ transformer model on VideoMix2M. We use a batch size of 3072 for the ViT-L/16 and ViT-H/16 models, and a batch

Table 1: *Pixels vs. Featurized Targets.* We ablate the effect of computing the prediction loss in feature space vs pixel space. All models are trained on VideoMix2M for 90K iterations with a batch size of 3072 using the multi-block prediction task. We examine downstream performance using a frozen backbone with attentive probing, and report top-1 accuracy using a single center view. We also examine end-to-end fine-tuning performance of the models on K400. Predicting in feature space provides a consistent improvement over pixel space prediction.

| | | *Frozen Evaluation* | | | *Fine-Tuning* |
|---|---|---|---|---|---|
| **Target** | **Arch.** | **K400** (16×1×1) | **SSv2** (16×1×1) | **IN1K** | **K400-ft** (16×5×3) |
| Pixels | ViT-L/16 | 68.6 | 66.0 | 73.3 | 85.4 |
| Features | ViT-L/16 | **73.7** | **66.2** | **74.8** | **85.6** |

Table 2: *Pretraining Data Distribution.* We pretrain all models for 90K iterations using a batch size of 3072, and evaluate downstream performance of the frozen backbones with an attentive probe using a single center view. Average performance across tasks increases with the pretraining dataset size.

| | | | *Frozen Evaluation* | | | |
|---|---|---|---|---|---|---|
| **Arch.** | **Data** | **#Samples** | **K400** (16×1×1) | **SSv2** (16×1×1) | **IN1K** | **Avg.** |
| | K710 | 700K | **75.8** | 63.2 | 73.7 | 70.9 |
| | K710+SSv2 | 900K | 72.9 | **67.4** | 72.8 | 71.0 |
| ViT-L/16 | K710+HT | 1900K | 74.5 | 64.2 | **74.8** | 71.1 |
| | 25% of VideoMix2M | 500K | 72.7 | 66.1 | 73.9 | 70.9 |
| | VideoMix2M | 2000K | 73.7 | 66.2 | **74.8** | **71.5** |
| | K710+SSv2 | 900K | **75.7** | 66.8 | 73.7 | 72.0 |
| ViT-H/16 | 25% of VideoMix2M | 500K | 73.5 | 67.6 | 75.2 | 72.1 |
| | VideoMix2M | 2000K | 74.0 | **68.5** | **75.9** | **72.8** |

size of 2400 for the ViT-H/16$_{384}$ model. Each model takes as input a video clip of 16 frames sampled with a frame-skip of 4, corresponding to roughly 3 second clips on average. The ViT-L/16 and ViT-H/16 process the video at a spatial resolution of 224, while the ViT-H/16$_{384}$ uses an input resolution of 384; cf. Appendix C.

**Evaluations.** Pretrained models are evaluated on downstream video and image tasks. On video tasks, we use a subset of the VideoGLUE benchmark (Yuan et al., 2023) to test for various capabilities; specifically, we investigate action recognition on Kinetics-400 (K400) (Kay et al., 2017), motion classification on Something-Something-v2 (SSv2) (Goyal et al., 2017), and action localization on AVA (Gu et al., 2018). Action classification on Kinetics evaluates the appearance-based understanding of the model, as many action classes in the dataset can be inferred from the presence of specific objects in the video (Sevilla-Lara et al., 2021). Motion classification on Something-Something-v2 evaluates the temporal understanding of the model, as action classes in the dataset are decoupled from the appearance/presence of specific objects in the video (Goyal et al., 2017). Finally, action localization on AVA evaluates the ability of the model to understand and localize motions in the video. We follow standard practice and report accuracy on K400 and SSv2 by sampling several spatial and temporal views. For static image tasks, we explore object recognition on ImageNet (Russakovsky et al., 2015), scene classification on Places205 (Zhou et al., 2014), and fine-grained recognition on iNaturalist 2021 (Van Horn et al., 2018).

## 4 What Matters for Learning Representations from Video?

In this section, we isolate the contributions of several design choices, including: a) the use of a feature prediction versus pixel prediction objective, b) the construction of the pretraining data distribution, c) the feature pooling strategy for leveraging the model's representations in downstream tasks, and d) the masking strategy, towards identifying: what to predict from what?

Table 3: *Average Pooling vs. Adaptive Pooling.* We pool the feature map output by the frozen V-JEPA encoder using an attentive probe, which is then fed into a linear classifier for downstream supervised tasks (K400 and SSv2). We evaluate two pooling strategies: 1) average pooling (Avg.), and attentive pooling (Att.). Results are reported using a single center view. Using adaptive pooling with a cross-attention layer leads to improvements of +17.3 points on K400 and +16.1 points on SSv2.

| | | *Frozen Evaluation* | | | |
| | | **K400** | | **SSv2** | |
| | | (16×1×1) | | (16×1×1) | |
| **Method** | **Arch.** | Avg. | Att. | Avg. | Att. |
| V-JEPA | ViT-L/16 | 56.7 | **73.7** | 50.1 | **66.2** |

### 4.1 Predicting Representations versus Pixels

We first ablate the effect of computing the prediction loss in representation space. We train a pair of ViT-L/16 models using either a V-JEPA feature prediction loss, or a mean-squared error loss with the normalized pixel values, as in masked autoencoders (He et al., 2021) and VideoMAE Tong et al. (2022). We perform a sweep over the learning rate and weight decay schedules for both approaches. All models are pretrained on VideoMix2M for 90K iterations with a batch size of 3072 using multi-block masking. We examine performance on Kinetics-400 (K400), Something-Something-v2 (SSv2), and ImageNet-1K (IN1K), using a frozen backbone with an attentive probe, and report top-1 accuracy using a single center view. We also examine end-to-end fine-tuning performance of the models on Kinetics-400.

Results of this comparison are reported in Table 1 and indicate that predicting in feature space provides a consistent performance improvement over pixel space prediction in both frozen evaluation of the video backbone, as well as end-to-end fine-tuning.

### 4.2 Pretraining Data Distribution

Next we study the impact of the pretraining data distribution in Table 2. Leveraging large scale datasets has been critical for enabling the surge of advancements in other modalities, such as text and images (Kaplan et al., 2020; Cherti et al., 2023). We investigate whether a similar trend holds for video data. To control for the possible confounding variable of compute budget, we pretrain all models in Table 2 for 90K iterations using a batch-size of 3072. We report downstream results on K400, SSv2, and IN1K using a frozen backbone with an attentive probe, and report top-1 accuracy using a single center view.

Table 2 shows that average performance across tasks monotonically increases as we increase the size of the pretraining dataset, but the best task-specific performance is obtained by independently selecting the pretraining data for each specific downstream task. For instance, the L/16 obtains its best SSv2 performance when pretrained on K710+SSv2, its best K400 performance when pretrained only on K710, and its best IN1K performance when pretrained only on K710+HT. The best average performance across all tasks is achieved by pretraining VideoMix2M, which combines all the data sources. Similarly, the H/16 pretrained on K710+SSv2 achieves a greater K400 score than the H/16 pretrained on VideoMix2M, however, the top performing H/16 on average is pretrained on VideoMix2M.

### 4.3 Evaluation: Attentive Probing

Next we explore the feature pooling strategy for applying the model's representations in downstream tasks. Specifically, when evaluating the frozen pretrained backbone on downstream tasks, we learn a cross-attention layer with a learnable query token. The output of the cross-attention layer is then added back to the query token (residual connection), and then fed into two-layer MLP with a single GeLU activation, followed by a LayerNorm, and finally a linear classifier. The attentive pooling strategy is similar in spirit to the typical use of a `[cls]` token in previous works to extract a single embedding from transformer representations in both vision (Oquab et al., 2023; Dosovitskiy et al., 2020) and language (Devlin et al., 2018).

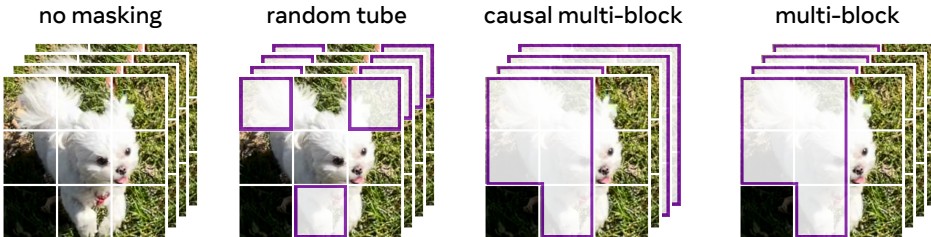

Figure 4: *Illustration of various video masking strategies for V-JEPA pretraining. Random Tube masking corresponds to masking a random number of tubelets in the video clip. Causal Multi-Block masking corresponds to masking out the last few frames of the video clip, as well as a random set of spatio-temporal blocks form the first few frames. Multi-Block masking corresponds to masking out a random set of spatio-temporal blocks form the entire video clip.*

Table 4: *Ablating Prediction Task.* Models are ViT-L/16 networks pretrained on K710 and SSv2 and evaluated with an attentive probe using a single center view. The region $x$ is sampled by masking spatio-temporal regions in the video; $y$ is the mask complement. **1) random-tube[r]:** $x$ is obtained by masking a fraction $r$ of tubes (spatial patches extended across the entire temporal duration) from the video, **2) causal multi-block[p]:** $x$ is restricted to the first $p$ frames of the 16-frame video, which are then masked with a random set of spatio-temporal blocks, **3) multi-block**: $x$ is obtained by masking a random set of spatio-temporal blocks from the entire video. Best performance obtained by using multiblock masking.

| | Frozen Evaluation | | |
|---|---|---|---|
| **Masking** | **K400** $(16\times1\times1)$ | **SSv2** $(16\times1\times1)$ | **IN1K** |
| random-tube[0.9] | 51.5 | 46.4 | 55.6 |
| causal multi-block[6] | 61.3 | 49.8 | 66.9 |
| causal multi-block[12] | 71.9 | 63.6 | 72.2 |
| multi-block | **72.9** | **67.4** | **72.8** |

In Table 3 we see that using adaptive pooling with a learnable cross-attention layer leads to a significant improvement of +17 points on K400 and +16.1 points on SSv2. Using an attentive-probe is also beneficial for other baseline models as reported in Appendix E.

### 4.4   Prediction Task: Predicting $y$ from $x$

We conduct an ablation on the masking strategy used in V-JEPA pretraining. We examine the following masking strategies: `random-tube[r]` in which $x$ is obtained by removing a random fraction $r$ of tubes (spatial patches extended across the entire temporal duration) from the video, `causal multi-block[p]` in which $x$ is restricted to the first $p$ frames of the 16-frame video, which are then masked with a random set of spatio-temporal blocks, and `multi-block` in which $x$ obtained by masking a random set of spatio-temporal blocks from the entire video. Spatio-temporal blocks are sampled using the parameters described in Section 4.4; an ablation on the size and quantity of masked spatio-temporal blocks is provided in Appendix E.3. All ablations are constructed by sampling two sets of masks for each input video clip. Figure 4 illustrates the conceptual differences between the various masking strategies.

Table 4 indicates that the best results are obtained by sampling $x$ using a *multi-block* strategy, wherein the network is forced to make predictions after removing large continuous blocks in the video. When $x$ is only sampled from the first few frames of the video, as in the *causal multi-block* strategy, we observe a decrease in downstream performances. Finally, the *random-tube* strategy, wherein 90% of the tubes in the video are randomly masked, leads to features of low-semantic quality when combined with our feature prediction objective.

Table 5: *Comparison with Pixel Prediction Methods.* We compare V-JEPA with OmniMAE (Girdhar et al., 2023), VideoMAE (Tong et al., 2022), and Hiera (Ryali et al., 2023), which leverage a pixel-reconstruction loss. All models are trained using a ViT-L architecture or a comparable Hiera-L. We evaluate the approaches on downstream image tasks (IN1K, Places205, iNat201) and video tasks (K400, SSv2, AVA) in both frozen evaluation (with a frozen backbone), and end-to-end fine-tuning. All models are evaluated at resolution 224. On K400 and SSv2 we follow the standard practice of reporting accuracy from several spatial and temporal views from the video. In frozen evaluation, V-JEPA outperforms the baselines on all downstream tasks, except ImageNet, where the model achieves 74.8% compared to 75.1% of an OmniMAE model trained directly on ImageNet. V-JEPA also achieves the best fine-tuning performance amongst all ViT-L models and matches the Hiera-L on SSv2. The V-JEPA results are achieved while processing significantly fewer examples during pretraining.

| Method | Arch. | #Samples Seen | Iter. | Frozen Evaluation w/ Att. Pooling | | | | | | Fine-Tuning | |
| | | | | K400 (16×8×3) | SSv2 (16×2×3) | AVA | IN1K | Places205 | iNat21 | K400-ft (16×5×3) | SSv2-ft (16×2×3) |
|---|---|---|---|---|---|---|---|---|---|---|---|
| *Methods pretrained using pixel prediction* | | | | | | | | | | | |
| OmniMAE | ViT-L/16 | 2400M | 1170K | 65.6 | 60.6 | 14.4 | **75.1** | 59.8 | 66.1 | 84.0 | 74.2 |
| VideoMAE | ViT-L/16 | 410M | 400K | 77.8 | 65.5 | 21.6 | 71.1 | 59.3 | 64.6 | 85.4 | 74.3 |
| Hiera | Hiera-L | 770M | 1500K | 75.5 | 64.2 | 15.8 | 68.9 | 58.5 | 56.9 | **87.3** | **75.1** |
| V-JEPA | ViT-L/16 | 270M | 90K | **80.8** | **69.5** | **25.6** | 74.8 | **60.3** | **67.8** | 85.6 | **75.1** |

Table 6: *Comparison with State-of-the-Art Models.* We compare V-JEPA with state-of-the-art baselines in frozen evaluation with an attentive probe on downstream image tasks (IN1K, Place205, iNat21) and video tasks (K400, SSv2, AVA). All models are evaluated at resolution 224, except I-JEPA$_{512}$ and V-JEPA$_{384}$ which are evaluated respectively at resolution 512 and 384. On K400 and SSv2 we follow the standard practice of reporting accuracy from several spatial and temporal views from the video. Compared to other video baselines, V-JEPA exhibits a consistent improvement across all downstream tasks. Compared to image-models that excel under the frozen evaluation, V-JEPA shows a significant performance improvement on tasks requiring motion understanding (+21 points on SSv2), and reduces the gap between video and image models on tasks requiring static appearance-based features.

| Method | Arch. | Params. | Data | Video Tasks | | | Image Tasks | | |
| | | | | K400 (16×8×3) | SSv2 (16×2×3) | AVA | IN1K | Places205 | iNat21 |
|---|---|---|---|---|---|---|---|---|---|
| *Methods pretrained on Images* | | | | | | | | | |
| I-JEPA | ViT-H/16$_{512}$ | 630M | IN22K | 79.7 | 50.0 | 19.8 | 84.4 | 66.5 | 85.7 |
| OpenCLIP | ViT-G/14 | 1800M | LAION | 81.8 | 34.8 | 23.2 | 85.3 | **70.2** | 83.6 |
| DINOv2 | ViT-g/14 | 1100M | LVD-142M | **83.4** | 50.6 | 24.3 | **86.2** | 68.4 | **88.8** |
| *Methods pretrained on Videos* | | | | | | | | | |
| MVD | ViT-L/16 | 200M | IN1K+K400 | 79.4 | 66.5 | 19.7 | 73.3 | 59.4 | 65.7 |
| OmniMAE | ViT-H/16 | 630M | IN1K+SSv2 | 71.4 | 65.4 | 16.0 | 76.3 | 60.6 | 72.4 |
| VideoMAE | ViT-H/16 | 630M | K400 | 79.8 | 66.2 | 20.7 | 72.3 | 59.1 | 65.5 |
| VideoMAEv2 | ViT-g/14 | 1100M | Un.Hybrid | 71.2 | 61.2 | 12.9 | 71.4 | 60.6 | 68.3 |
| Hiera | Hiera-H | 670M | K400 | 77.0 | 64.7 | 17.5 | 71.4 | 59.5 | 61.7 |
| V-JEPA | ViT-L/16 | 200M | | 80.8 | 69.5 | 25.6 | 74.8 | 60.3 | 67.8 |
| | ViT-H/16 | 630M | VideoMix2M | **82.0** | 71.4 | **25.8** | 75.9 | 61.7 | 67.9 |
| | ViT-H/16$_{384}$ | 630M | | 81.9 | **72.2** | 25.0 | **77.4** | **62.8** | **72.6** |

# 5 Comparison with Prior Work

In Section 5.1, we investigate the impact of feature prediction by comparing V-JEPA with video approaches that rely on pixel prediction, while using a similar architecture for all baselines. Subsequently, in Section 5.2, we remove the architectural constraint and report the best performance across architectures for self-supervised video and image pretraining approaches. Finally, we explore the label-efficiency of V-JEPA relative to other self-supervised video pretraining approaches in Section 5.3. We further detail the evaluation setup in Appendix D.

## 5.1 Comparison with Pixel Prediction

To investigate the effectiveness of feature prediction pretraining, we first compare V-JEPA to video masked modeling models relying on a pixel prediction loss. We control for the possible confounding factor of model architecture by evaluating all models using either a ViT-L/16 encoder, or a Hiera-L encoder, which has a

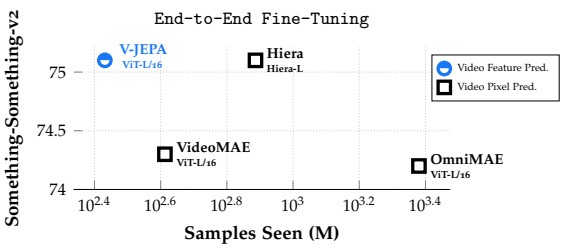

(a) *SSv2 fine-tuning vs. Samples Seen.*  (b) *SSv2 frozen-evaluation vs. Pretraining Time.*

Figure 5: Efficiency of pretraining. **(a)** We report SSv2 fine-tuning for V-JEPA and pixel-reconstruction baselines using a ViT-L/16 or Hiera-L architecture. V-JEPA outperforms all pixel-reconstruction methods using a ViT-L/16 and matches the Hiera-L performance while seeing significantly less samples during pretraining. **(b)** Wallclock times for all methods are measured on a single GPU with a batch size of 10 clips, using the official codebases for VideoMAE and VideoMAEv2, and linearly extrapolated assuming a global batch size of 2400 samples. However, note that the SSv2 accuracies of video pixel prediction methods are actually obtained with small batch sizes and significantly longer training schedules. V-JEPA outperforms pixel-reconstruction methods while training significantly faster.

similar number of parameters. For the pixel prediction baselines we consider VideoMAE (Tong et al., 2022; Wang et al., 2023a), which trains vision transformer autoencoders exclusively on video, Hiera (Ryali et al., 2023), which trains a hierarchical transformer autoencoder on video, and OmniMAE (Girdhar et al., 2023), which trains a vision transformer autoencoder on static images and video simultaneously.

Table 5 examines both frozen evaluation with an attentive probe on downstream video and image tasks, as well as end-to-end fine-tuning. In frozen evaluation, V-JEPA outperforms the baselines on all downstream tasks, except ImageNet, where we achieve 74.8% compared to 75.1% of an OmniMAE model trained directly on ImageNet; hence, V-JEPA achieves comparable ImageNet performance despite only pretraining on video.

Under the fine-tuning protocol, V-JEPA also achieves the best performance of any model trained with a ViT-L/16, and matches the performance of the Hiera-L on SSv2, which benefits from a hierachical prior (Ryali et al., 2023). The V-JEPA models achieve this result while processing significantly fewer samples during pretraining (Figure 5a), demonstrating the efficiency of feature prediction as a learning principle.

## 5.2 Comparison with State-of-the-Art

Next, in Table 6, we inspect how the V-JEPA models pretrained on video stack up next to the largest state-of-the-art self-supervised image and video models when freezing the backbone encoder and training an attentive probe on top. Our image pretrained baselines include OpenCLIP (Cherti et al., 2023), DINOv2 (Oquab et al., 2023), and I-JEPA (Assran et al., 2023). The OpenCLIP model is trained with a contrastive image-text alignment objective, DINOv2 and I-JEPA are trained with self-supervision. These models are known to excel in their frozen-evaluation performance (Oquab et al., 2023); i.e., their ability to produce visual features that can be applied to many downstream tasks simultaneously, without end-to-end fine-tuning, and thus provide highly competitive baselines. Our video pretrained baselines include VideoMAE (Tong et al., 2022), OmniMAE (Girdhar et al., 2023), Hiera (Ryali et al., 2023), VideoMAEv2 (Wang et al., 2023a), and MVD (Wang et al., 2023b). The OpenCLIP, DINOv2 and VideoMAEv2 models are parameterized as Giant/Gigantic vision transformer architectures containing over 1B parameters trained on large-scale image or video datasets.

**Comparison with video models.**    Compared to large-scale video baselines, the V-JEPA models outperform all previous models on every downstream video and image tasks by a notable margin (see Table 6). Our H/16 model outperforms the largest publicly available VideoMAE, VideoMAEv2, OmniMAE, MVD, and Hiera models by at least +5 points in motion understanding (Something-Something-v2), +2 points in action recognition (Kinetics-400), +5 points on action detection (AVA), +1 point on object recognition (ImageNet-1K), +2 points in scene recognition (Places205), and +0.2 points on fine-grained recognition

Table 7: **Finetuning results.** We evaluate a V-JEPA model with the finetuning protocol on the K400 and SSv2 datasets using 16 frames per clip and multi-view fusion (5×3 or 2×3) for inference. The **#Samples Seen** entry corresponds to the number of video clips processed during pretraining, which is larger than the size of the pretraining dataset for multi-epoch training. We compare V-JEPA with different video self-supervised learning approaches. We report the VideoMAEv2 results without instruction-turning for consistency with the other approaches. V-JEPA obtains competitive performance using the finetuning protocol.

| Method | Arch. | Pretraining Data | #Samples Seen | K400 (16×5×3) | SSv2 (16×2×3) |
|---|---|---|---|---|---|
| VideoMAEv1 | ViT-L/16 | K400\|SSv2 | 380M\|410M | 85.4 | 74.3 |
| | ViT-H/16 | K400\|SSv2 | 380M\|410M | 86.6 | 74.8 |
| VideoMAEv2 | ViT-H/16 | Un.Hybrid | 1600M | 86.9 | 76.8 |
| MVD | ViT-L/16 | K400+IN1K | 2400M | 86.4 | 76.7 |
| | ViT-H/16 | K400+IN1K | 2400M | **87.2** | **77.3** |
| V-JEPA | ViT-L/16 | VideoMix2M | 270M | 85.6 | 75.1 |
| | ViT-H/16 | VideoMix2M | 270M | 86.6 | 77.0 |

(iNaturalist). Moreover, when comparing pretraining wallclock time in Figure 5b, we see that V-JEPA achieves this performance with a roughly 2× speedup compared to the large pixel prediction models.

**Comparison with image models.** On tasks that require a fine-grained understanding of motion (Something-Something-v2), the V-JEPA models provide a major improvement (over +21 points) compared to large-scale image baselines, such as DINOv2, OpenCLIP, and I-JEPA. Self-supervised pretraining from videos allows to model dynamic concepts that are not easily learned from static image datasets. Similarly, we observe that the V-JEPA models outperform image-based pretraining on action localization.

On Kinetics-400, we find image models to perform well; e.g., while DINOv2 (Oquab et al., 2023) previously reported 78.4% on K400 with a linear probe, we improve the frozen evaluation of the g/14 model to 83.4% by using an attentive probe. In this case, our H/16 model achieves 82.0% top-1 accuracy. It is worth noting that the label for many Kinetics videos can be inferred using appearance-based cues, without requiring an understanding of motion (Sevilla-Lara et al., 2021).

The V-JEPA models narrow the gap with image models on image classification tasks. In particular, V-JEPA achieves a score of 77.4% on ImageNet using a one-layer attentive probe, which can be further improved to **77.9**% using a two-layer attentive probe. More generally, we hypothesize that the datasets used to train V-JEPA and other video models are too constrained and lack the visual diversity of the internet-scale pretraining data used by the images models; as such, there is value in focusing future work on building diverse publicly available video datasets.

**Finetuning** In Table 7, we evaluate V-JEPA by finetuning (separately) on K400 and SSv2. We compare V-JEPA with VideoMAEv2 (Wang et al., 2023a), VideoMAE (Tong et al., 2022) and MVD (Wang et al., 2023b) using a ViT-L/16 and a ViT-H/16 architecture. V-JEPA obtains competitive performance using a finetuning protocol. With a ViT-H/16 architecture, V-JEPA outperforms VideoMAE by +1.2% and VideoMAEv2 by +0.3% on the SSv2 dataset, while obtaining comparable performance on K400. V-JEPA also obtains performance similar to MVD on the SSv2 dataset. The MVD model achieves the best performance across models on the K400 dataset, and is trained using an additional image dataset (ImageNet1K), in contrast to the other methods in the table, which only use video data. Additionally MVD requires the processing of significantly more samples during pretraining due to the cost of training the teacher encoder networks in a pre-pre-training stage.

## 5.3 Label-efficiency

We examine the label-efficiency of V-JEPA compared to other self-supervised video models by measuring the ability of the pretrained backbones to adapt to downstream tasks with few labels. Specifically, we investigate the performance of the frozen models on Kinetics-400 and Something-Something-v2 as we vary the percentage of labeled examples from each dataset available for training the attentive probe. We train the probes in several low-shot settings: using either 5% of the train set, 10%, or 50%, and take 3 random splits in each

Table 8: *Low-Shot Frozen Evaluation.* Comparing V-JEPA to other video models in frozen evaluation on Kinetics-400 and Something-Something-v2 as we vary the percentage of labeled examples from each dataset available for training the attentive probe. We train the probes in several low-shot settings: using either 5% of the train set, 10%, or 50%, and take 3 random splits in each setting to obtain more robust metrics, resulting in 9 different evaluation experiments for each model. We report the mean performances and standard deviation using the K400 and SSv2 validation sets. V-JEPA is more label-efficient than other models; specifically, decreasing the available number of labeled examples from each class increases the performance gap between V-JEPA and the baselines.

| | | Frozen Evaluation | | | | | |
| | | K400 (16×8×3) | | | SSv2 (16×2×3) | | |
| Method | Arch. | 5% (∼29 samples per class) | 10% (∼58 samples per class) | 50% (∼287 samples per class) | 5% (∼48 samples per class) | 10% (∼96 samples per class) | 50% (∼440 samples per class) |
|---|---|---|---|---|---|---|---|
| MVD | ViT-L/16 | $62.6 \pm 0.2$ | $68.3 \pm 0.2$ | $77.2 \pm 0.3$ | $42.9 \pm 0.8$ | $49.5 \pm 0.6$ | $61.0 \pm 0.2$ |
| VideoMAE | ViT-H/16 | $62.3 \pm 0.3$ | $68.5 \pm 0.2$ | $78.2 \pm 0.1$ | $41.4 \pm 0.8$ | $48.1 \pm 0.2$ | $60.5 \pm 0.4$ |
| VideoMAEv2 | ViT-g/14 | $37.0 \pm 0.3$ | $48.8 \pm 0.4$ | $67.8 \pm 0.1$ | $28.0 \pm 1.0$ | $37.3 \pm 0.3$ | $54.0 \pm 0.3$ |
| V-JEPA | ViT-H/16 | $67.0 \pm 0.2$ | $72.1 \pm 0.1$ | $80.2 \pm 0.2$ | $51.9 \pm 0.3$ | $57.5 \pm 0.4$ | $67.3 \pm 0.2$ |
| | ViT-H/16$_{384}$ | $\mathbf{68.2 \pm 0.2}$ | $\mathbf{72.8 \pm 0.2}$ | $\mathbf{80.6 \pm 0.2}$ | $\mathbf{54.0 \pm 0.2}$ | $\mathbf{59.3 \pm 0.5}$ | $\mathbf{67.9 \pm 0.2}$ |

setting to obtain more robust metrics, resulting in 9 different evaluation experiments for each model. Table 8 reports the mean performances and standard deviation using the K400 and SSv2 validation sets.

We find V-JEPA to be more label-efficient than other self-supervised video models: decreasing the available number of labeled examples for training the attentive probe results in an increase in the performance gap between V-JEPA and the other models. In particular, the performance of the largest V-JEPA model on K400 drops by 12% to 68.2% top-1 when we reduce the number of labeled examples by a factor of 10× (from roughly 287 examples per class to 29 examples per class). By contrast, VideoMAEv2 drops by 30% to 37.0% top-1, VideoMAE drops by 15.9% to 62.3% top-1, and MVD drops by 14.6% to 62.6% top-1.

Similar observations hold on SSv2. The performance of the largest V-JEPA model on SSv2 drops by 13.9% to 54.0% top-1 when we reduce the number of labeled examples by a factor of 10× (from roughly 440 examples per class to 48 examples per class). By contrast, VideoMAEv2 drops by 26% to 28.0% top-1, VideoMAE drops by 19.1% to 41.4% top-1, and MVD drops by 18.1% to 42.9% top-1.

## 6 Evaluating the Predictor

Next, we seek to qualitatively inspect the V-JEPA models. Recall that the predictor network in V-JEPA predicts the representations of a masked spatio-temporal region $y$ from a visible region $x$, given the positional information of the masked regions (see Section 3). To qualitatively investigate the grounding of the feature-space predictions, we freeze the pretrained encoder and predictor networks and train a conditional diffusion decoder to map the V-JEPA predictions to interpretable pixels (Bordes et al., 2021). Notably, the decoder is only fed the representations predicted for the missing regions of the video, and does not have access to the unmasked regions of the video (see Figure 6a).

Given a masked video, we use the V-JEPA pretrained models to predict the representations of the missing regions, and then use the decoder to project the representations to pixel space. Figure 6b shows decoder outputs for various random seeds. Qualities that are common across samples represent information that is contained in the predictor representation.

Figure 6b shows that the V-JEPA feature predictions are indeed grounded, and exhibit spatio-temporal consistency with the unmasked regions of the video. Specifically, the samples in Figure 6b show that the V-JEPA predictor correctly captures positional uncertainty and produces a variety of visual objects at various locations with consistent motion. Some of the samples also demonstrate an understanding of object-permanence, as the visual objects remain consistent after partial occlusion.

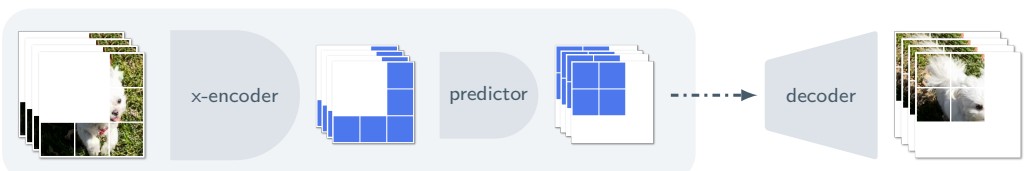

(a) **Visualization Methodology.** We train a conditional diffusion model to decode the V-JEPA feature-space predictions to interpretable pixels; the pretrained V-JEPA encoder and predictor networks are kept frozen in this process. The decoder is only fed the representations predicted for the missing regions of the video, and does not have access to the unmasked regions of the video.

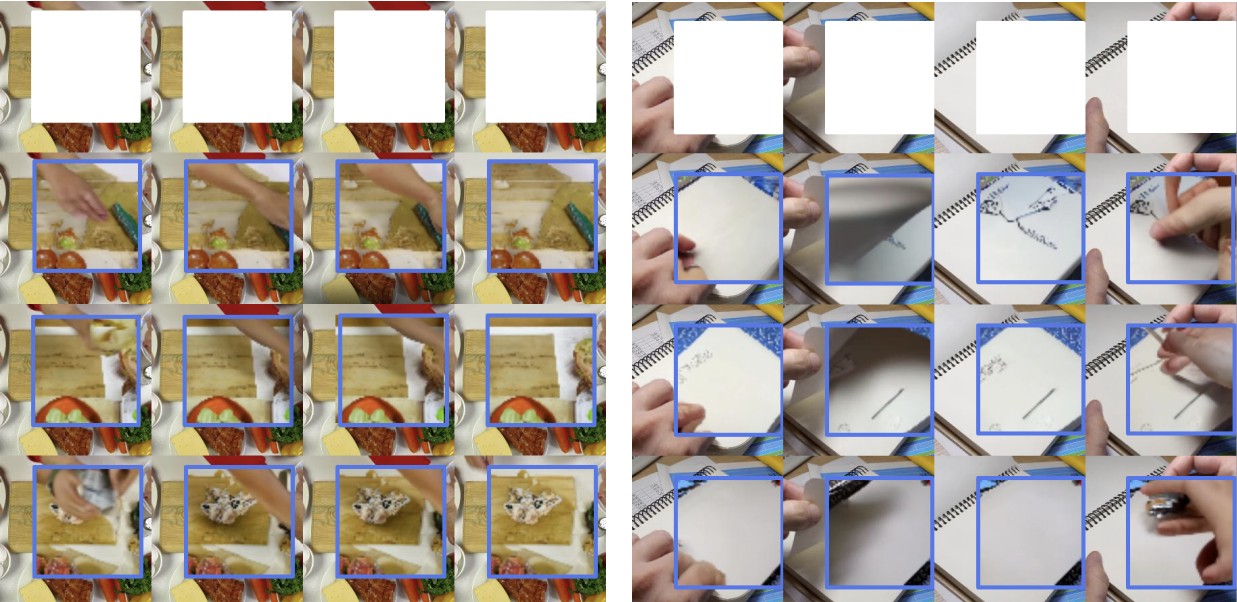

(b) **Visualizations.** *First Row:* Masked videos used as input to the V-JEPA models (a pretrained ViT-H/16 encoder and its corresponding predictor network). *Other rows:* Bounding boxes contain various samples from the decoder overlayed on the original video. V-JEPA is not a generative model and the decoder does not have access to the context (first row), so we do not expect samples to exactly match the input. This experiment qualitatively illustrates what information is encoded and predicted by V-JEPA. In particular, characteristics that are common across samples represent information that is encoded in the V-JEPA predictions. V-JEPA generates predictions that are spatially and temporally coherent with unmask region of the video. The predictions also capture consistent motion through time.

Figure 6: *Qualitative Analysis.* Offline visualizations of the V-JEPA feature-space predictions.

# 7   Limitations

While the focus of this work is on exploring the effectiveness of feature prediction as a stand-alone objective for unsupervised learning for video, in this section, we highlight current limitations of our instantiation of this learning objective, V-JEPA. Firstly, V-JEPA performance is sensitive to the masking strategy, as illustrated in Section 4.4. In future work, we plan to investigate whether scaling the dataset size and pretraining computation make the approach more robust to this design choice. Secondly, the V-JEPA instantiation relies on vision transformers as video encoders, which limits the ability to process long or high-resolution video, because the computational complexity scales quadratically with the input size. We plan to investigate hierarchical architectures to enable the processing of larger videos. Finally, the V-JEPA instantiation relies on bootstrapping to prevent representation collapse (Grill et al., 2020). Despite initial results (Tian et al., 2021), understanding the learning mechanism behind this principle remains an open question.

## 8 Conclusion

In this work, we explored the effectiveness of feature prediction as a stand-alone objective for unsupervised learning from video and introduced V-JEPA, a collection of vision models trained solely using a self-supervised feature prediction objective. The V-JEPA models demonstrate the ability to solve various downstream image and video tasks without adaption of the model parameters, and outperform previous video representation learning approaches in frozen evaluation on action recognition, spatio-temporal action detection, and image classification tasks. Additionally, we show that pretraining V-JEPA on videos is particularly effective for solving downstream tasks requiring fine-grained motion understanding, while large-scale image models trained on internet scale datasets fall short on such tasks. Finally, we empirically observed that V-JEPA models are label-efficient learners, and exhibit good performance on downstream tasks, even when only few labeled examples are available.

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

## A  Extended Related Works

We first review approaches for learning visual perception from static images before discussing strategies for learning from video.

### Weakly-Supervised Learning from Static Images

One family of approaches for learning visual perception from static images trains a visual encoder to predict the representations of text captions often found accompanying images from the Web, as in CLIP (Radford et al., 2021) or CoCa (Yu et al., 2022). The largest open source CLIP model to date, numbering 2B parameters and trained on over 2B web-scraped images (Cherti et al., 2023), demonstrates impressive performance on a wide range of downstream image and video tasks. Notably, this is achieved using only the light-weight adaptation of task-specific heads, also referred to as frozen-evaluation, and does not require expensive end-to-end fine-tuning of the pretrained model.

### Self-Supervised Learning from Static Images

Other approaches for learning from static images leverage unsupervised objectives. Initial works on self-supervised approaches are based on sparse coding or hand-crafted pretext tasks, such as colorization (Larsson et al., 2016; 2017), rotation prediction (Gidaris et al., 2020), and jigsaws (Noroozi & Favaro, 2016). More recent approaches leverage invariance-based objectives by training a visual encoder to be invariant to hand-crafted image transformations (Wu et al., 2018; Chen et al., 2020).

Another family of methods learn representations using denoising autoencoders (Vincent et al., 2008); image inpainting is one popular instantiation of this idea (Pathak et al., 2016). More recently, masked autoencoders (He et al., 2021) train an encoder-decoder transformer to predict missing pixels of a masked image. Follow-up work addresses the indeterminism of pixel reconstruction by exploring instantiations of masked image modeling in latent space (Baevski et al., 2022b; Assran et al., 2023; Baevski et al., 2022a). These approaches can be seen as applications of the predictive feature principle in the image modality.

There are also various methods that combine both masked image modeling and invariance criteria to learn visual representations from static images, such as iBOT (Zhou et al., 2021) and DINOv2 (Zhou et al., 2021; Oquab et al., 2023), the latter is currently the most competitive instantiation of self-supervised learning with static images, scaled to a model with over 1.1B parameters trained on a curated dataset of 142M images.

### Weakly-Supervised Learning from Videos

One family of approaches for learning visual perception from videos relies on weakly-supervised guidance from closed captioning, often computed from an ASR transcription of audio data accompanying internet videos. For instance, VideoBERT (Sun et al., 2019; Xu et al., 2021) trains a video encoder to predict masked spans in the textual closed captions. Similarly, VideoCLIP (Xu et al., 2021) trains a video encoder to predict the representation of video captions computed by a text encoder. Follow-up work such as MERLOT (Zellers et al., 2022), VATT (Akbari et al., 2021), and InternVideo (Wang et al., 2022) extended VideoCLIP by incorporating additional unsupervised objectives.

**Self-Supervised Learning from Videos**

Similar to unsupervised learning from images, a family of unsupervised video representation learning approaches enforces a spatio-temporal representation of a video clip to be invariant to hand-crafted spatio-temporal data augmentations (Parthasarathy et al., 2022). However, one obvious insight is that the temporal ordering of visual information in video can provide implicit supervision. Indeed, this insight is the key insight leveraged by many works on unsupervised video learning. Towards leveraging temporal information as supervision, some approaches train a visual encoder by predicting the temporal ordering of frames (Xu et al., 2019; Lee et al., 2017). Other approaches seek to predict low-level motion vectors computed from optical flow (Pintea et al., 2014), or to predict mixing pixels in video frames, using either a frame-interpolation objective (Kalluri et al., 2023) or a denoising autoencoder (Tong et al., 2022; Feichtenhofer et al., 2022; Wang et al., 2023a).

## B    Extended Description of V-JEPA

In this section, we provide an in-depth description of our approach V-JEPA that is illustrated in Figure 3.

**Input.**    Unless stated otherwise, during during pretraining, we always randomly sample a clip of 16 frames from each input video with a temporal stride of 4 between sampled frames. An input video clip therefore covers 64 frames in total, or roughly 2 seconds of a given video running at 30 frames per second. We then resize the video's spatial dimensions to $224 \times 224$, resulting in an overall shape of $16 \times 224 \times 224 \times 3$ for the entire clip. Since ViT networks process a 1D sequence of tokens, we must convert an input video clip into a 1D token sequence. To do so, we apply a 3D convolution comprising $d$ filters of size $2 \times 16 \times 16$ with a temporal stride of 2 and a spatial stride of 16, resulting in a tensor of shape $8 \times 14 \times 14 \times d$. Next we add absolute 3D sin-cos positional embeddings to the spatio-temporal feature map and flatten it, resulting in a 1D token sequence of shape $1568 \times d$. This process is demonstrated in Figure 7.

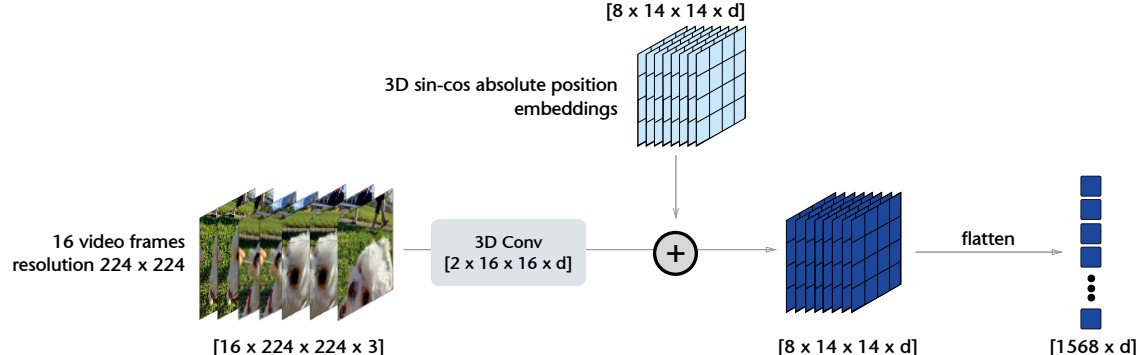

Figure 7: **V-JEPA** training operates on a video clip flattened into a sequence of tokens. To convert a video clip of size $16 \times 224 \times 224 \times 3$ into a 1D token sequence, we apply a 3D convolution comprising $d$ filters of size $2 \times 16 \times 16$ with a temporal stride of 2 and a spatial stride of 16, resulting in a tensor of shape $8 \times 14 \times 14 \times d$. Next we add absolute 3D sin-cos positional embeddings to the spatio-temporal feature map and flatten it, resulting in a 1D token sequence of shape $1568 \times d$.

**V-JEPA.**    We sample both a video clip, and a video mask in each iteration. We denote a video clip represented as a 1D token sequence of length $L = 1568$ by $x_L = (x_1, \ldots, x_L)$. Similarly, given a mask of $M < L$ patches, leaving $N = L - M$ patches unmasked, we denote the indices of masked patches by $(i_1, \ldots, i_M)$ and its complement (the indices of unmasked patches) by $(j_1, \ldots, j_N)$.

*Computing the x-representations.* To compute the V-JEPA loss, we first produce the $x$-representations by masking the video clip and feeding it into the $x$-encoder; we denote the masked video by $x_N = (x_{j_1}, \ldots, x_{j_N})$. Applying the $x$-encoder $E_\theta(\cdot)$ to the masked clip gives a sequence of patch representations, denoted as $h_N = E_\theta(x_N) = (h_{j_1}, \ldots, h_{j_N})$.

*Predicting the target.* Next, the V-JEPA predictor network $P_\phi(\cdot, \cdot)$ takes as input the tokens produced by the $x$-encoder and predicts the missing regions in the video clip, which are specified by a set of learnable mask tokens. Specifically, the mask tokens are parameterized as the sum of a shared learnable vector and an absolute 3D sin-cos positional embedding, denoted by $m_M = (m_{i_1}, \ldots, m_{i_M})$. The output of the predictor is thus given by, $\hat{s}_M = P_\phi(z_N, m_M) = (\hat{s}_{i_1}, \ldots, \hat{s}_{i_M})$, corresponding to a $d$-dimensional output for each of the $M$ masked patches.

*Computing the y-representations.* Finally to compute the prediction targets, the entire unmasked video clip is processed by the $y$-encoder to obtain a set of target representations, denoted by $s_L = \overline{E}_\theta(x_L) = (s_1, \ldots, s_L)$. The V-JEPA loss is now computed as

$$\text{Loss} = \frac{1}{M} \sum_{k \in (i_1, \ldots, i_M)} \|\hat{s}_k - s_k\|_1, \tag{2}$$

which is simply the average $L_1$ distance between the output of the predictor and the $y$-encoder. We then compute a gradient update with respect to the parameters of the $x$-encoder, $\theta$, and the predictor, $\phi$, and subsequently update the parameters of the $y$-encoder as an exponential moving average of the context encoder weights (Polyak average).

**Multi-Mask Prediction.** To increase the efficiency of V-JEPA, we use a multi-masking strategy (Caron et al., 2020; Baevski et al., 2022a), which enables us to amortize the cost of the target computation. As mentioned in Section 3, for a given video clip, we sample 2 different masks: short-range and long-range. While we need to forward propagate the $x$-encoder and predictor separately for each mask, we only need to compute the $y$-representation once. Specifically, we forward propagate the unmasked video through the $y$-encoder once to compute the target representations, and forward propagate through the $x$-encoder and predictor twice, one time using long-range masks, and a second time using short-range masks; since roughly 90% of the video clip is masked, the two forward passes through the $x$-encoder are reasonably efficient and amortize the cost of target computation.

## C    Pretraining details

In section, we report V-JEPA pretraining details. Table 9 summarizes the main hyperparameters used during pretraining.

**Architectures.** We use Vision Transformer (Dosovitskiy et al., 2020) (ViT) architectures for the $x$-encoder and $y$-encoder. We train three V-JEPA encoders: a ViT-L/$16_{224}$, a ViT-H/$16_{224}$ and a ViT-H/$16_{384}$. All three encoders take as input a short video clip of 16 frames with a temporal stride of 4 between consecutive frames. The subscripts, 224 and 384, indicate the spatial resolution of the video clip. V-JEPA flattens the video clip into a sequence of non-overlapping spatio-temporal patches of size $16 \times 16 \times 2$ (see Figure 7). For all three models, the predictor is designed as a narrow ViT architecture, consisting of 12 transformer blocks with an embedding dimension of 384. For simplicity, we keep the number of self-attention heads in the predictor equal to that of the backbone used for the context-encoder/target-encoder. V-JEPA is pretrained *without* using a `[cls]` token.

**Optimization.** We use AdamW (Loshchilov & Hutter, 2017) to optimize the $x$-encoder and predictor weights. The ViT-L/$16_{224}$ and ViT-H/$16_{224}$ models use a batch size of 3072 while the ViT-H/$16_{384}$ uses a batch size of 2400. Models are trained for a total of 90,000 iterations. The learning rate is linearly increased from $2 \times 10^{-4}$ to $6.25 \times 10^{-4}$ during the first $12,000$ iterations of pretraining, and decayed to $10^{-6}$ following a cosine schedule. Weight-decay is also linearly increased from 0.04 to 0.4 throughout pretraining. The $y$-encoder weights are initialized identically to the $x$-encoder, and subsequently updated as an exponential moving average (EMA) (Tarvainen & Valpola, 2017) of the $x$-encoder weights using a momentum value which starts at 0.998 and is linearly increased to 1.0 during training (Caron et al., 2021; Assran et al., 2022). We scale all hyper-parameter schedules 25% beyond the actual training schedule. Specifically, the learning rate schedule, weight-decay schedule, and EMA schedule are computed assuming a training length of 112,500

Table 9: **pretraining hyper-parameters for V-JEPA.**

| Hyper-parameter | ViT-L/16$_{224}$ | ViT-H/16$_{224}$ | ViT-H/16$_{384}$ |
|---|---|---|---|
| *data* | | | |
| datasets | VideoMix2M | VideoMix2M | VideoMix2M |
| resolution | 224 | 224 | 384 |
| num_frames | 16 | 16 | 16 |
| temporal_stride | 4 | 4 | 4 |
| horizontal_flip | true | true | true |
| random_resize_scale | (0.3, 1.0) | (0.3, 1.0) | (0.3, 1.0) |
| random_resize_aspect_ratio | (0.75, 1.35) | (0.75, 1.35) | (0.75, 1.35) |
| *masking* | | | |
| block_aspect_ratio | (0.75, 1.5) | (0.75, 1.5) | (0.75, 1.5) |
| shortrange_mask_num_blocks | 8 | 8 | 8 |
| shortrange_mask_spatial_scale | 0.15 | 0.15 | 0.15 |
| longrange_mask_num_blocks | 2 | 2 | 2 |
| longrange_mask_spatial_scale | 0.7 | 0.7 | 0.7 |
| *optimization* | | | |
| batch_size | 3072 | 3072 | 2400 |
| total_number_of_iterations | 90000 | 90000 | 90000 |
| warmup_iterations | 12000 | 12000 | 12000 |
| lr | 6.25e-4 | $6.25\times10^{-4}$ | $6.25\times10^{-4}$ |
| start_lr | $2\times10^{-4}$ | $2\times10^{-4}$ | $2\times10^{-4}$ |
| final_lr | $1\times10^{-6}$ | $1\times10^{-6}$ | $1\times10^{-6}$ |
| start_momentum | 0.998 | 0.998 | 0.998 |
| final_momentum | 1.0 | 1.0 | 1.0 |
| start_weight_decay | 0.04 | 0.04 | 0.04 |
| final_weight_decay | 0.4 | 0.4 | 0.4 |
| scheduler_scale_factor | 1.25 | 1.25 | 1.25 |
| *architecture* | | | |
| patch_size | 16 | 16 | 16 |
| tubelet_size | 2 | 2 | 2 |
| pred_depth | 12 | 12 | 12 |
| pred_embed_dim | 384 | 384 | 384 |
| *hardware* | | | |
| dtype | bfloat16 | bfloat16 | bfloat16 |
| accelerator | A100 80G | A100 80G | A100 80G |

Table 10: **Frozen Evaluation hyper-parameters.**

| Hyper-parameter | K400 | SSv2 | IN1K | Place205 | iNat21 |
|---|---|---|---|---|---|
| *data* | | | | | |
| num_clips | 8 | 1 | N.A. | N.A. | N.A. |
| num_frames | 16 | 16 | N.A. | N.A. | N.A. |
| temporal_stride | 4 | 4 | N.A. | N.A. | N.A. |
| horizontal_flip | true | true | true | true | true |
| random_resize_scale | (0.08, 1.0) | (0.08, 1.0) | (0.08, 1.0) | (0.08, 1.0) | (0.08, 1.0) |
| random_resize_aspect_ratio | (0.75, 1.33) | (0.75, 1.33) | (0.75, 1.33) | (0.75, 1.33) | (0.75, 1.33) |
| auto_augment | false | false | true | true | true |
| *optimization* | | | | | |
| batch_size | 256 | 256 | 1024 | 1024 | 1024 |
| epochs | 20 | 20 | 20 | 20 | 20 |
| lr | 1e-3 | 1e-3 | 1e-3 | 1e-3 | 1e-3 |
| final_lr | 0 | 0 | 0 | 0 | 0 |
| weight_decay | 0.01 | 0.01 | 0.01 | 0.01 | 0.01 |

iterations, although we only train our model for 90,000 iterations. We found the last 25% of the default scheduler period to update hyper-parameters too aggressively, and simply truncating the schedulers improved performance. To find the optimal values $v$ for the learning rate (scheduled from $2 \times 10^{-4}$ to $v$), weight decay (scheduled from $v$ to 0.4), and momentum (scheduled from $v$ to 1.0), we performed a grid search with the following values, learning rate: $3e-3, 1e-3, 7.5e-4, 6.25e-4, 5e-4, 3.75e-4, 2e-4$, weight decay: $0.04, 0.004, 0.0004, 0.00004$, momentum: $0.99, 0.995, 0.996, 0.997, 0.998, 0.999$.

**Masking.** As described in Section 3, we propose a 3D Multi-Block masking strategy. We use two types of masks: short-range masks, where we take the union of 8 randomly sampled target blocks with a spatial scale of 0.15, and long-range masks, where we take the union of 2 randomly sampled target blocks with a spatial scale of 0.7. In both cases, the aspect ratio for all sampled blocks is randomly chosen in the range $(0.75, 1.5)$.

# D   Evaluation details

## D.1   Frozen classification

**Attentive Probing.** Given an input video, $x_L$, the V-JEPA target encoder $\overline{E}_\theta(\cdot)$ outputs a sequence of $L$ tokens, $E_\theta(x_L) = (s_1, \ldots, s_L)$, where $s_i \in \mathbb{R}^d$. To pool this sequence of tokens into a single feature vector, we apply a lightweight non-linear cross-attention block which replaces the self-attention operation of a transformer block with cross attention. Specifically, cross-attention performs the following computation:

$$\sum_{i=1}^{L} \frac{\exp(q^\top \mathbf{W_k} s_i)}{\sum_j \exp(q^\top \mathbf{W_k} s_j)} \mathbf{W_v} s_i,$$

where $\mathbf{W_k}, \mathbf{W_v} \in \mathbf{R^{d \times d}}$ are the key and value matrices, and $q \in R^d$ is a learnable query token. The output of the cross-attention is then added back to the query token (residual connection), and then fed into two-layer MLP with a single GeLU activation, followed by a LayerNorm, and finally a linear classifier. The parameters of the cross-attention block are jointly learned with that of the linear classifier for the downstream task, while the encoder parameters are kept frozen. Note that in practice, we actually use an attentive probe with 12 heads, each of dimension 12. In Appendix E we show that baselines benefit from the attentive probing protocol.

**Optimization.** For all the tasks, we use AdamW optimizer with a cosine scheduler (no warmup) that decays the learning rate from 0.001 to 0. We use a fixed weight-decay of 0.01 and apply simple data augmentations (random resized crops and horizontal flips) during training of the attentive probe, except on image tasks, where we apply AutoAugment (Dogus Cubuk et al., 2019). Table 10 reports the hyperparameters for each downstream evaluation.

Table 11: **Frozen Detection hyper-parameters.**

| Hyper-parameter | ViT-L/16 | ViT-H/16 |
|---|---|---|
| out_layers | [18, 20, 22, 24] | [26, 28, 30, 32] |
| batch_size | 64 | 64 |
| epochs | 30 | 30 |
| opt | AdamW | AdamW |
| opt_eps | 0.00000001 | 0.00000001 |
| momentum | 0.9 | 0.9 |
| weight_decay | 0.05 | 0.05 |
| lr | 0.0001 | 0.0001 |
| warmup_lr | 0.000001 | 0.000001 |
| min_lr | 0.000001 | 0.000001 |
| warmup_epochs | 2 | 2 |
| warmup_steps | 1 | 1 |

**Extension to multiple clips.** Unless stated otherwise, our attentive probe takes 8 clips of 16 frames as input on Kinetics, and 2 clips of 16 frames on Something-Somethingv2 to increase the temporal coverage of the video. Specifically, we first divide a video in 8 (or 2) equal-length temporal segments, and sample 1 clip at random per segment. The video encoder $\overline{E}_\theta$ processes each clip separately and produces a clip-level feature map. The feature maps for each clip are then concatenated together and fed to the attentive probe. At test time, we average the prediction of 3 spatial views following standard practice in video classification.

**Application of video models to images.** To evaluate the video models on image tasks, we simply duplicate input images to generate still video clips of 16 frames. We perform this duplication operation simply for convenience in evaluation of the video models, however we find this step to be unnecessary in general. Given a video tokenizer implemented as a 3D-conv with a temporal stride of 2, it is sufficient to simply duplicate the image into a 2 frame video clip. This would result in the same number of input tokens as that produced by a static image model with a 2D-conv tokenizer.

**Application of image models to videos.** To evaluate image models such as DINOv2 and OpenCLIP on video tasks, we simply process each frame independently with the image encoder to produce a frame-level feature map. The feature maps for each frame are then concatenated and fed to the attentive probe, just as we do with the clip-level feature maps when evaluating video models.

### D.2 Frozen detection

We evaluate our model on the AVA (Gu et al., 2018) spatio-temporal localization of human actions dataset, containing 211k training and 57k validation video segments. We follow the experimental protocol of (Feichtenhofer et al., 2021), and use precomputed masks from a pretrained Faster-RCNN adapted to videos, which uses a ResNeXt-101-FPN backbone and is pretrained on ImageNet and COCO. We train a linear classifier on top of the *frozen* V-JEPA features to classify the extracted regions of interest and report mean Average Precision (mAP) on the 60 most common classes. Hyper-parameters are provided in Table 11. Our frozen features are obtained by concatenating the last layer of the transformer encoder with three intermediate layers. We use a batch size of 64 and pretrain for 30 epochs with AdamW using a learning rate of 0.0001 with 2 epochs of warmup and a weight decay of 0.05.

### D.3 Finetuning

Following Tong et al. (2022), we finetune a linear layer on top of our model, using a layer decay schema and mixup as the data augmentation pipeline. We provide all hyper-parameters for both K400 and SSv2 in Table 12.

Table 12: **Finetuning Evaluation hyper-parameters.**

| Hyper-parameter | K400 | | SSv2 | |
|---|---|---|---|---|
| *data* | | | | |
| num_segments | | 1 | | |
| num_frames | | 16 | | |
| sampling_rate | | 4 | | |
| resolution | | 224 | | |
| *model* | | | | |
| model_name | ViT-L/16 | ViT-H/16 | ViT-L/16 | ViT-H/16 |
| drop_path | 0.1 | 0.2 | 0.2 | 0.2 |
| head_drop_rate | 0. | 0. | 0.5 | 0.5 |
| *optimization* | | | | |
| batch_size | 256 | 1024 | 256 | 256 |
| epochs | 35 | 25 | 15 | 15 |
| opt | | adamw | | |
| opt_eps | | 0.00000001 | | |
| momentum | | 0.9 | | |
| weight_decay | | 0.05 | | |
| lr | 0.002 | 0.0005 | 0.0005 | 0.0005 |
| layer_decay | 0.75 | 0.75 | 0.75 | 0.75 |
| warmup_lr | 1e-6 | 1e-8 | 1e-6 | 1e-6 |
| min_lr | 1e-6 | 1e-5 | 1.5e-4 | 1.5e-3 |
| warmup_epochs | | 5 | | |
| *augmentations* | | | | |
| color_jitter | | 0.4 | | |
| horizontal_flip | True | True | False | False |
| num_sample | | 2 | | |
| aa | | rand-m7-n4-mstd0.5-inc1 | | |
| smoothing | | 0.1 | | |
| train_interpolation | | bicubic | | |
| test_num_segment | 5 | 5 | 2 | 2 |
| test_num_crop | 3 | 3 | 3 | 3 |
| *erase* | | | | |
| prob | | 0.25 | | |
| mode | | pixel | | |
| count | | 1 | | |
| split | | False | | |
| *mixup* | | | | |
| mixup | | 0.8 | | |
| cutmix | | 1.0 | | |
| mixup_prob | | 1.0 | | |
| mixup_switch_prob | | 0.5 | | |
| mixup_mode | | batch | | |

# E    Extra Results

## E.1    Frozen Evaluation.

Table 13: **Linear vs. Attentive Probe Evaluation for V-JEPA and VideoMAE.** We evaluate the effect of linear (Lin.) and attentive (Att.) probing when adapting V-JEPA to the K400 ($16 \times 5 \times 3$) and SSv2 ($16 \times 2 \times 2$) tasks. V-JEPA and VideoMAE benefit from using a non-linear attentive probe.

| Method | Arch. | K400 Lin. | K400 Att. | SSv2 Lin. | SSv2 Att. |
|---|---|---|---|---|---|
| VideoMAE | ViT-L/16 | 52.5 | 77.8 | 41.3 | 61.2 |
| V-JEPA | ViT-L/16 | 56.7 | **80.8** | 50.1 | **69.5** |

Table 14: **Linear vs. Attentive Probe Evaluation for DINOv2 and OpenCLIP.** We evaluate the effect of linear (Lin.) and attentive probing (Att.) when adapting DINOv2 and OpenCLIP. Image-baselines benefit from using an attentive probing strategy. Results shown in gray are reported from the linear probe evaluation in Oquab et al. (2023).

| Method | Arch. | K400 Lin. | K400 Att. | SSv2 Lin. | SSv2 Att. | IN1K Lin. | IN1K Att. | Place205 Lin. | Place205 Att. | iNat21 Lin. | iNat21 Att. |
|---|---|---|---|---|---|---|---|---|---|---|---|
| DINOv2 | ViT-g/14 | 78.4 | 83.4 | 38.3 | 50.0 | 86.5 | 86.2 | 67.5 | 68.4 | 85.7 | 88.8 |
| OpenCLIP | ViT-G/14 | 78.3 | 81.8 | 35.8 | 34.8 | 86.2 | 85.3 | 69.8 | 70.2 | 76.0 | 83.6 |

**Linear vs. Attentive probe**    Table 13 shows that V-JEPA and VideoMAE benefit from using a non-linear attentive probe and multiple clips on the K400 and SSv2 downstream tasks. Additionally, Table 14 shows that attentive probing leads to better performance on average for DINOv2 and OpenCLIP models. Since attentive probing and multiclips eval improves the performance of all models, we use it as our default protocol in frozen evaluation.

Table 15: **Temporal Coverage on Kinetics-400.** We evaluate the effect of temporal coverage on K400. We train an attentive probe on K400 using either 1 clip ($\approx 2$ seconds of a video) or 8 clips ($\approx 16$ seconds of a video). To sample $N$ clips, we first divide a video in $N$ equal-length temporal segments and sample one clip at random per segment. The video encoder processes each clip in parallel and all the encoder output tokens are concatenated at the input of the attentive probe. Increasing the temporal coverage from 1 clip per video to 8 clips significantly improves the performance for both our VideoMAE baseline and V-JEPA.

| Method | Arch. | 1 Clip | 8 Clips |
|---|---|---|---|
| VideoMAE | ViT-L/16 | 69.4 | 77.8 |
| V-JEPA | ViT-L/16 | 73.7 | 80.9 |

**One Clip vs Multiple clips.**    We examine the impact of changing the temporal coverage of a model during downstream evaluation on K400 action classification. In Table 15, we evaluate VideoMAE and V-JEPA models using an attentive probe with access to either the feature map of 1 clip randomly sampled from the video, or the concatenated feature map of 8 clips randomly sampled from the video. To sample 8 clips from a video, we first divide the video into 8 equal length temporal segments, and sample 1 clip at random from each segment. A single clip corresponds to $\approx 2$ seconds of a video on average, while 8 clips correspond to $\approx 16$ seconds. The video encoders processes each clip separately to produce a clip-level feature map, which are then concatenated at the input to the attentive probe.

Increasing the temporal coverage from 1 clip per video to 8 clips improves the performance of both V-JEPA and VideoMAE on K400 action classification. We therefore use the multiclip attentive probing setup as our default evaluation pipeline.

## E.2    Sample Efficiency of pretraining

We compare the sample efficiency of pretraining various state-of-the-art image and video models. Specifically, we look at the number of samples (image or video clips) processed by the network during pretraining, which

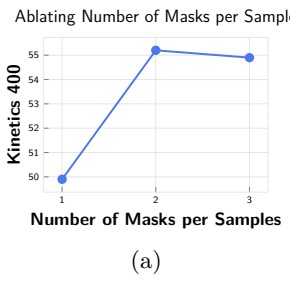
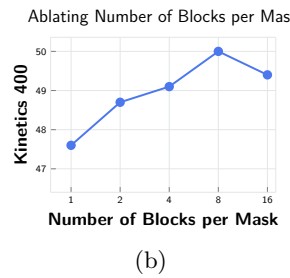
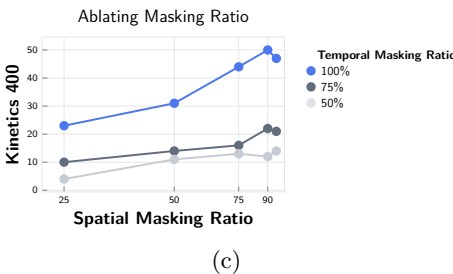

(a)             (b)             (c)

Figure 8: **Masking Strategy Ablation.** Evaluating a linear probe on a ViT-B/16 pretrained with V-JEPA on K400 under various 3D Multi-Block masking settings. We examine the impact of **(a)** sampling several masks per video, **(b)** varying the number of blocks in a mask, and **(c)** varying the average spatial and temporal masking ratio. A temporal masking ratio of 100% extends the spatial mask across all the frames in the clip. We find it important to maintain a high spatial and temporal masking ratio during pretraining.

is larger than the size of the pretraining dataset for multi-epoch training. Notably, our results with V-JEPA are obtained while processing an order of magnitude fewer samples than previous methods, and notably two orders of magnitude fewer samples than OpenCLIP. We believe that further investment towards improving the video pretraining data distribution could lead to substantial gains in downstream image and video tasks.

Table 16: **Sample efficiency.** We compare the sample efficiency of pretraining various state-of-the-art image and video models. The **#Samples Seen** entry corresponds to the number of samples (image or video clips) processed by the network during pretraining, which is larger than the size of the pretraining dataset for multi-epoch training. The V-JEPA results in this paper are obtained while processing an order of magnitude fewer samples than previous methods.

| Method | Arch. | Data | #Samples Seen |
|---|---|---|---|
| OpenCLIP | ViT-G/14 | LAION-2B | 39000M |
| DINOv2 | ViT-g/14 | LVD 142M | 1900M |
| VideoMAEv2 | ViT-g/14 | UnlabeledHybrid | 1600M |
| V-JEPA | ViT-H/16$_{384}$ | VideoMix2M | 210M |

### E.3 Masking Strategy

An important component of the V-JEPA pretraining strategy is the 3D clip masking strategy. In this section, we detail 26 ablation experiments exploring different masks. For all the experiments, we pretrain a ViT-B/16 pretrained on K400. Figure 8 presents a summary of those results.

Figure 8c shows the effect of changing the spatial and temporal masking ratio. Figure 8b ablates the number of sampled blocks used to construct the masks given a fixed effective masking ratio of 90%. Finally, in Figure 8a we examine our multi-masking strategy and find that sampling two masks for each clip (long-range and short-range) to be more effective than sampling just a single mask for each clip.

In Figure 8c, we explore different average spatial and temporal masking ratio, i.e. the spatial/temporal ratio of the area that is covered by a mask on average for a clip. Recall that each mask is constructed by sampling several (possibly overlapping) blocks and taking their union. We change the average spatial or temporal masking ratio by changing a block spatial or temporal size, as well as the overall number of blocks. We found that low spatial or temporal coverage results in a trivial prediction task, which degrades downstream performance. Based on those results, we sample masks that remove roughly 90% of the frame and extend along the entire temporal dimension of the clip by default.

In Figure 8b , we explore different block size given an effective spatial masking ratio of 90% and temporal ratio of 100%. We keep the masking ratio approximately constant by changing the block size and the number of block at the same time. We find that sampling several blocks to perform better than sampling a single large block. Figure 9 visually illustrates the effect of sampling several smaller blocks to construct a mask.

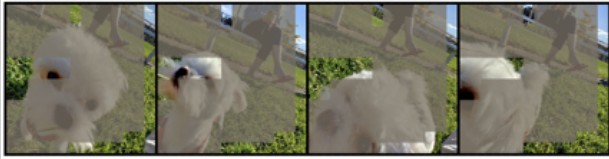

(a) Num. Blocks: 8, Spatial Block Size: $32 \times 32$

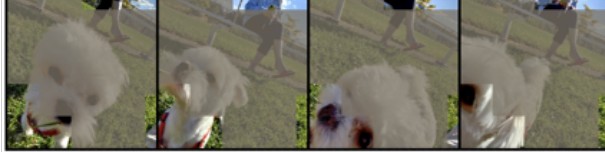

(b) Num. Blocks: 4, Spatial Block Size: $80 \times 80$

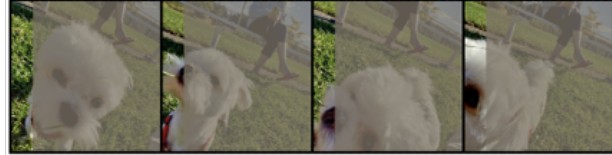

(c) Num. Blocks: 2, Spatial Block Size: $160 \times 160$

Figure 9: Illustration of mask with number of blocks and block size. Each mask is constructed by sampling several (possibly overlapping) blocks and taking their union.

In Figure 8a, we explore the effect of sampling various number of masks per sample. We find that sampling two masks for each clip, with different spatial block sizes for each, to be more effective than sampling just a single mask. We hypothesize that this masking strategy induces complementary tasks. In our experiment, we use this as our default masks sampling.

## F  Predictor Visualization Details

To visualize V-JEPA predictions in pixel space, we use the RCDM framework Bordes et al. (2021). Given an input pair $(x, y)$, we train a generative diffusion model to reconstruct $y$ from $P_\phi(E_\theta(x))$[1], a prediction of $y$ given $x$ in representation space, and a noisy version of $y$: $\hat{y} := y + \epsilon$, where $\epsilon$ is an additive noise vector.

Specifically, we train a decoder network $D_\omega$ to minimize the loss function $\|D_\omega(\hat{y}, P_\phi(E_\theta(x))) - \epsilon\|_2^2$, where $x$ is a masked video and $y$ is a sequence of 16 frames at resolution $64 \times 64$ corresponding to the masked region. The parameters of the V-JEPA predictor $P_\phi$ and encoder $E_\theta$ are kept frozen when training the decoder. We use a V-JEPA ViT-H/16$_{224}$ model.

$D_\omega$ is parameterized as a 3D-Unet composed by 4 downsampling blocks with 3 residual layers per downsampling block. The base channel dimension of $D_\omega$ is set to 192. We train our decoder for 450,000 iterations. All other hyperparameters (architecture, optimization, noise schedule...) are identical to the one used in RCDM.

After training the decoder, one can subsequently feed the prediction vector of unseen test videos into the decoder along with various random noise vectors to generate several pixel-level visualizations of the predictions.

---

[1]we drop the mask tokens in the notation for simplicity.

