# OpenReview forum: "Revisiting Feature Prediction for Learning Visual Representations from Video"
_TMLR — Accepted by TMLR_

### Review · Reviewer_Lsst · 2024-05-08

**Summary Of Contributions:**

Summary: The paper explores feature prediction as a stand-alone objective for unsupervised learning from video. It introduces V-JEPA (Video Joint-Embedding Predictive Architecture), a collection of vision models trained purely on a feature prediction objective without using pretrained image encoders, text, negative examples, reconstruction, or other sources of supervision (which is one of the most interesting elements of this paper, arguably). The models are trained on 2 million videos collected from public datasets and evaluated on downstream image and video tasks. The results show that learning by predicting video features lead to versatile visual representations that perform well on motion and appearance-based tasks without adapting the model's parameters. \
In terms of the main minor and major contributions: \
1.	The paper introduces V-JEPA, a relatively simple method based on feature prediction for unsupervised learning from video (largely inspired by Assran's and LeCun's I-JEPA work – to me very promising directions in general) \
2.	They show the effectiveness of feature prediction as a stand-alone objective for learning visual representations from video without other sources of supervision. \
3.	They evaluated V-JEPA on various downstream image and video tasks, showing its ability to learn versatile representations that perform well on motion and appearance-based tasks. \
4.	They run several comparisons between V-JEPA and other self-supervised video representation learning approaches (including Dino) and showing its advantages in terms of performance and efficiency. \
5.	The authors presented work demonstrating the effectiveness of design choices such as feature prediction vs. pixel prediction, pretraining data distribution, feature pooling strategy, and masking strategy. \
6.	They provided some qualitative analysis of the learned representations through some visualisations.

**Audience:**

Yes

**Claims And Evidence:**

Yes

**Requested Changes:**

1.	Please include a dedicated section or subsection discussing the potential limitations or failure modes of the proposed approach, and outlining future research directions.
2.	Please consider adding some fine tuning bits and hyperparemeters in the main paper.
3.	Please include an ablation study or discussion on the impact of different architectures and hyperparameters on the performance of V-JEPA (I am not too invested in this, and to some extent some details are included in the appendix)

**Strengths And Weaknesses:**

Strengths:
I think I have already outlined the main strengths of the paper above in the contributions but to summarise: \
1.	The method is conceptually simple and involves feature prediction, without relying on other sources of supervision or auxiliary objectives. \
2.	The authors have done a very comprehensive evaluation on various downstream image and video tasks, demonstrating the versatility of the learned representations. \
3.	They have provided some ablations studies (and a very extensive appendix) \
4.	Comparison with state-of-the-art self-supervised video and image representation approaches \
5.	Qualitative analysis through visualisation, providing insights into the learned representations. \

Weaknesses: \
1.	Limited discussion on the potential limitations or failure modes of the proposed approach; more related ablation studies would have helped. To be fair the authors have added a very comprehensive appendix, but I do not think it helps having a massive appendix with key results. \
2.	The paper primarily focuses on the evaluation of frozen backbones, with limited analysis of end-to-end fine-tuning performance (appendix). \
3.	The paper is relatively light on elaborating and expanding on hyperparameters on the performance

Some questions \

1.	Could you provide more insights into the potential limitations or failure modes of the V-JEPA approach?
2.	what about the fine tuning results? Wouldn't it more suitable to have them in the main paper?
3.	I appreciate that you have added the hyperparameters in the appendix, but similarly to point 2, maybe some of the details could have been included in the main paper.
4.	Given the limited size of the pretraining dataset (VideoMix2M), how do you expect the performance of V-JEPA to scale with larger and more diverse video datasets? in the grand scheme of things, is this dataset large (e.g. akin to how we consider datasets like ImageNet.

---

> ### Author Response · Authors · 2024-06-07
>
> Thank you for your review.
>
> **Potential limitations.** We added a paragraph in the conclusion to discuss current limitations of the approach. Specifically we see three main limitations of V-JEPA: 1) V-JEPA performances are sensitive to the masking strategy, as illustrated in the ablations.  In future work, we will investigate if scaling (pretraining data size, pretraining computation budget) decreases sensitivity to these design choices. 2) Our approach relies on vision transformers as video encoders, which limits the ability to process long or high-resolution video as the computational complexity scales quadratically with the input size. We plan to investigate hierarchical architectures to enable the processing of bigger videos. 3) Our approach relies on bootstrapping to learn representations. While there is plenty of empirical evidence demonstrating that bootstrapping is a compelling approach, it still remains an open question to fully understand why this type of approach is effective and we need more research in this area.
>
> **Finetuning results.** Following your suggestion, we move Table 15 to the main body of the paper to provide better visibility to the finetuning results.
>
> **Hyper-parameter ablations.** We provide in the paper an ablation on the masking strategy, which was the main driver for achieving good performance with the method. We ablate the number of masks per sample, the masking ratio, and the size and number of the masked blocks.
>
> Regarding optimization parameters, we conducted a wide grid search on learning rate, momentum, and weight decay. We added in Appendix C, paragraph Optimization, a description of the specific values we tried.
>
> Regarding the encoder architecture, we simply follow standard approaches and use ViT-L and ViT-H backbones commonly used in video processing, using a patch size of 16, a tubelet size of 2, and sampling 16-frame clips with a frame-skip of 4, as standard in the literature. Regarding the predictor architecture, we follow I-JEPA and use a deep but shallow predictor network, and fixed the number of layers to 12 and embedding size to 384 for all experiments.
>
>
> **Dataset size.** While VideoMix2M contains 2M samples, we qualitatively observed that its visual diversity was more constrained than a typical image dataset such as ImageNet, which contains a broader range of visual concepts. We think that increasing the visual diversity of the video pre training dataset is a very promising avenue to further improve the quality of visual features learned from videos.
>
> Diversity can take various forms, object and scene diversity could be improved by taking inspiration from the DINOv2 pretraining dataset distribution, but would require gathering videos that cover this distribution. Beyond object diversity, increasing motion and interaction diversity would allow us to better learn the dynamics of objects as opposed to simply learning their semantics.

---

> ### Comment · Reviewer_Lsst · 2024-06-09
>
> Potential limitations:  Thank you for reflecting on that - I see that you have added a new section on limitations which is great
>
> Finetuning: Thank you for moving the table to the main body; I personally think it is now more complete
>
> Hyper-parameter ablations: that is clear now - thank you.
>
> Dataset size: Thank you

---

> > ### Author Response · Authors · 2024-06-11
> >
> > Thank you for your helpful and comprehensive review.

---

### Review · Reviewer_bA7B · 2024-05-12

**Summary Of Contributions:**

This work proposes a method for self-supervised representation learning via a combination of masking and bootstrapped representation prediction. The proposed method involves masking spatial regions from a video, tokenizing the results, processing them with a vision transformer to obtain representations of each token, and then training a predictor to reconstruct the (moving-average) representations of masked-out patches from the kept patches. The authors conduct a wide set of experiments and ablations to understand the performance of their approach, and find that it outperforms previous techniques on motion-understanding tasks and is competitive with previous approaches for content-understanding.

**Audience:**

Yes

**Broader Impact Concerns:**

No broader impact concerns.

**Claims And Evidence:**

Yes

**Requested Changes:**

## Important issues

Before recommending acceptance, I'd like the authors to address a few clarity issues:

### Attentive probing
Section 4.3 states that the representations may not be linearly separable because the prediction objective is unnormalized. However, I don't understand why being linearly separable should be related to the normalization of the objective. Linear separability is preserved under rescaling representations by a constant (or, indeed, any invertable linear operation). The authors must mean something else by "unnormalized", but I don't follow what it is.

The authors also state that the reason for using attentive probing instead of averaging is because averaging is a linear operation. But this motivation also doesn't make sense to me. Averaging seems to have more to do with aggregating information than it does with linear separability, so I don't see why the linearity of averaging is relevant.

Finally, the attentive probing strategy proposed here seems somewhat similar to existing methods used for transformer representations, using e.g. a `[class]` (or `[CLS]`) token to extract an embedding (as in the [ViT](https://arxiv.org/abs/2010.11929) and [BERT](https://arxiv.org/abs/1810.04805v2) papers). I think this connection should be discussed more.

### Masking strategy details

I found the description of the masking strategy to be somewhat difficult to follow. I also didn't fully understand the difference between the "random-tube" and "multi-block" strategies; both of these use multiple "spatial patches extended across the entire temporal duration", right? What is the difference between a block and a tube?

I think a figure comparing the different masking strategies in Section 4.4 would be useful.

### What is $z$?

Figure 2 in Section 3 states that there is an additional variable $z$ that "provides the predictor with information about the transformation that computes $y$ from $x$." But I couldn't find a definition of $z$ for the proposed model. What is it?

I notice that the appendix uses $z$ to refer to the encoded representation of $x$, but I don't think that's the same thing, since Figure 2 shows $z$ as a separate conditioning signal. (This notation should probably be modified to be more consistent.)

---

## Other suggestions

I also think the paper could be improved by expanding on the below points, although I do not think these are necessary for acceptance.

### Joint relationship between dataset size and diversity in Table 2 / Section 4.2
It seems that Table 2 and Section 4.2 vary the dataset size and composition jointly. The largest dataset is also the most diverse dataset, which includes three data sources. As such, the "average" gains could be due to the increased diversity, rather than size alone.

I think it would be interesting to separate how much of the gains come from the diversity of the data, and how much from the size. These could be disentangled by subsampling all of the datasets, e.g. taking 25% or 50% of VideoMix2M, and seeing how that affects performance.

### Dependence of bootstrapping procedure on the initial representation

The authors justify the V-JEPA bootstrapping procedure (in Section 3.1) by arguing that their objective leads the encoder to capture information about the video in order to minimize the conditional median-absolute-deviation of the masked tokens given the learned encoding.

However, it seems that this argument relies on the initial target variable $Y$ capturing important features of the input. If $Y$ is invariant to some important property of the input, the encoder does not seem to have a reason to learn it.

This indicates that the initialization of the encoder would be important for the method to succeed. I'd appreciate some more discussion on this. How do you initialize the encoder to avoid missing important features of the input?

### L1 loss

I'm curious about the use of L1 regression in the objective. The authors mention stability; was this more stable relative to the L2 norm? (The L1 norm depends on the specific basis vector used, so I also wonder if this leads to any sort of sparsity or feature specialization.)

### Masking before/after the encoder
The paper mentions that masking is done at the input when processing $x$, but at the output when processing $y$. It seems like this could be a potential distribution shift for the model, since (if I understand correctly) gradients are only calculated with respect to the embeddings of $x$, never for $y$. Do you observe any issues with distribution shift or length generalization due to feeding in more unmasked tokens?

**Strengths And Weaknesses:**

Strengths:

- **S1.** The paper is overall clearly written and well explained. The authors do a good job motivating their approach and putting it in the context of prior work.
- **S2.** The proposed approach seems to work well. The learned representations outperform or are competitive with prior work, and the "attentive probing" evaluation protocol also boosts performance (of their method and also baselines).
- **S3.** The authors study a number of interesting questions through their experiments and ablations, such as the performance of representation prediction v.s. pixel prediction, and the label efficiency of different learned representations.
- **S4.** All of the claims of the paper are precisely stated and seem well-supported by the evidence.

Weaknesses:

- **W1.** A few aspects of the motivation for attentive probing were difficult to understand, and the relationship between attentive probing and previous representation learning techniques could be better explained.
- **W2.** Some details of the masking strategy and conditioning signals could be explained more precisely.
- **W3.** The results in Table 2 and Section 4.2 seem to vary both dataset size and dataset composition together, which makes it hard to disentangle their effects.

---

> ### Author Response · Authors · 2024-06-07
>
> Thank you for your review.
>
> **Attentive pooling.** Indeed, we intended a different meaning here. The loss is unnormalized in the sense that it does not linearly discriminate between other video clips in a batch (the denominator in contrastive methods). We will remove these two sentences, as they are inconsequential for the paper. The benefit of attentive pooling is an empirical finding, supported by Table 12 and 13 in the appendix. We agree that an attentive probe is similar in spirit to the use of a [CLS] token to aggregate information from transformer representations and have clarified the connection in the main text; thank you for pointing this out. The main differences with the [CLS] token are that: 1) we add the query token at the end of the encoder while a [CLS] is added at the input, and 2) we perform a cross-attention operation, i.e., the query token does not self-attend to itself.
>
> **Masking strategy.** Thank you for the suggestions, we added a figure to better illustrate the masking strategy in Section 4.4.
>
> **What is z.** z is simply the spatio-temporal position of the patch we are trying to predict. In practice, we parametrize z as the sum of a learnable mask token and a 3D sincos position embedding. You are correct, the notation is inconsistent with Figure 2. We have updated the notation in the appendix.
>
> **Dataset size vs diversity.**
> As suggested, we trained a ViT-L/16 on a random 25% subset of VM2M for the same 90K optimization steps; the results are summarized below. Even with an identical computational budget, we see a positively monotonic trend on all tasks when controlling for the data distribution.
>
> *ViT-L/16 (90K optimization steps)*
> | | K400 | SSv2 | IN1k |
> | -- | -- | -- | -- |
> | **25%** | 72.7 | 66.1 | 73.9 |
> | **100%** | 73.7 | 66.2 | 74.8 |
> | | | |
>
> *ViT-H/16 (90K optimization steps)*
> | | K400 | SSv2 | IN1k |
> | -- | -- | -- | -- |
> | **25%** | 73.5 | 67.6 | 75.2 |
> | **100%** | 74.0 | 68.5 | 75.9
> | | | |
>
> **Dependance on initial representation.** The argument on median absolute deviation does not rely on the initial target variable Y capturing important features at initialization, as Y is dependent on the encoder parameters $\theta$. By leading the encoder to capture more diverse features, the targets are bootstrapped to also capture more diverse features through the moving average encoder. More generally, there is significant empirical evidence demonstrating that bootstrapping is a competitive approach to learn representations [BYOL, IJEPA, SimSiam, Data2Vec]. However, better understanding the learning mechanism behind this principle remains largely an open question.
>
> **L1 loss.** We use L1 loss simply because it penalizes outlier samples less than a quadratic loss. We did not investigate the effect of L1 on the sparsity of the representation or feature specialization.
>
> **Masking before/after encoder.** Yes, you are correct in your understanding. We did not observe any issue with respect to distribution shift or sequence length generalization. Additionally, this design choice was ablated in the I-JEPA paper and was shown to improve the quality of the representation (see Table 11 in appendix of the I-JEPA paper).

---

> > ### Comment · Reviewer_bA7B · 2024-06-08
> >
> > Thanks for your reply and for the changes to the submission.
> >
> > **Attentive pooling.** This makes sense and addresses my concern.
> >
> > **Masking strategy.** Thanks for adding the new figure, which I agree greatly clarifies the differences between the masking strategies. Am I correct in understanding that the difference between a tube and a block is that a tube is always the size of a single token/patch in image space, whereas a block usually covers multiple tokens?
> >
> > **What is z.** I see, thanks for clarifying. I think it would be worthwhile to state which part of the input is $z$ in Section 3.3 and Figure 3, to help readers match it with Figure 2 and the start of Section 3. If I understand correctly, $z$ is the "sequence of learnable mask tokens with
> > positional embeddings indicating the spatio-temporal positions of the $y$ tokens" described in Section 3.3, shown as the red tokens in the "Concatenate mask tokens" step of Figure 3?
> >
> > **Dataset size vs diversity.** Thanks for these new results. I'd suggest also including them also as a row in Table 2.
> >
> > **Dependance on initial representation.** My point is that it at least requires capturing *some* important features, e.g. if $Y$ was a constant zero at initialization then there is no incentive to learn anything at all. More realistically, I could imagine there being some features that never get bootstrapped into the targets because they aren't useful for predicting the previous targets. (But I agree this is probably out of scope of this work given that the same question applies to most previous bootstrapping approaches as well.)

---

> > > ### Author Response · Authors · 2024-06-11
> > >
> > > **Masking strategy.** Yes, this understanding is correct.
> > >
> > > **What is z.** Yes, your understanding of $z$ is also correct. Thank you for the suggestion, we clarified which part of the input is $z$ in Section 3.3.
> > >
> > > **Dataset size vs diversity.**  We have added the results in Table 2.
> > >
> > > **Dependance on initial representation.**  Yes, it is correct that a zero initialization for bootstrapping approaches would be problematic; we agree that more research is needed to fully understand how bootstrapping-based objectives learn useful representations, and have mentioned this in Section 7, and hope to see more work in this area.
> > >
> > > Thank you again for your careful review and all your detailed comments!

---

### Review · Reviewer_qEpG · 2024-05-29

**Summary Of Contributions:**

The paper explores the effectiveness of joint-embedding predictive architecture (JEPA) in self-supervised learning on videos. By evaluating the model on both image and video tasks, the authors demonstrate that the model trained with feature prediction is better in both compute efficiency and label efficiency when transfer to downstream tasks — V-JEPA model achieves competitive results by using attentive probing while pixel reconstruction models (e.g., MAE) needs fine-tuning. In addition, the fewer supervised labels used in transfer learning, the larger the gap is between V-JEPA and other pixel reconstruction models.

**Audience:**

Yes

**Broader Impact Concerns:**

No broader impact concerns.

**Claims And Evidence:**

Yes

**Requested Changes:**

1. **Section 3.2**: It is unclear that how long-range and short-range mask are combined.  Do you take the union of the two mask to be applied to a single sample, or do you apply them separately to a duplicated sample and keep both of them in the same batch? Please specify it more explicitly.
2. **Section 6**: *“We freeze the pretrained encoder and predictor networks and train a conditional diffusion decoder to map the V-JEPA predictions to interpretable pixels”*, please cite the conditional diffusion model you used, and add the implementation details to the Appendix.
3. **Table 5 and Table 6**: Models are trained on different number of data. It will be good to add another line for V-JEPA trained on K400 only to have a fair comparison with models like Hiera and VideoMAE.
4. **Table 5**: It’d be more complete to add VideoMAEv2 into the table as a stronger competitor than VideoMAE. The VideoMAEv2 fine-tuning results are only shown in Table 15, better to have it to Table 5 as well.

**Strengths And Weaknesses:**

### **Strength**:
1. The first paper to explore joint-embedding predictive architecture in video domain, it is a well-studied extension to I-JEPA [Assran et al, 2023). It studies different masking and probing strategies that were not used in the image domain and achieves good results on a wide range of image and video tasks.
2. Compared to other self-supervised methods, the proposed V-JEPA has the  multiple advantages: a) Its training schedule is  relatively shorter. b) it achieves competitive results by attentive probing without the need to fine-tune the whole model, which saves the compute when transfer to downstream tasks. c) It achieves much better results on down-stream tasks in a low data-regime
3. The method is clearly introduced and the findings are supported by thorough experiments and comparisons.

### **Weakness**:
1. It is not well discussed that why V-JEPA has to use attentive-probing to achieve good results on classification while  I-JEPA has good performance on classification just by linear probing.
2. When the JEPA method is applied to video, apart from the tube masking, there seems to be very little adaptation to the method which considers or makes use of features of videos. Hence, it is hard to know whether the method is able to solve problems poses only in videos, for instance, motion estimation, object tracking etc. For example, I-JEPA and V-JEPA achieves similar results on Kinetics, and its performance on AVA is not much better than DinoV2.  The good performance on SSV2 seems to indicate its better motion modeling ability. However, it would be good to support it by evaluating on  a) classification on more challenging datasets (e.g., EpicKitchens, Ego4D), b) more other non-semantic tasks like tracking or spatio-temporal localization.
3. It is not clear how much the model will benefit from more data. In Table 2, the average improvement by scaling the data from 700K to 2M is only 0.6.
4. There are some comparisons missing or experiments details need to be clarified -- see the ‘requested changes’ section for details.

---

> ### Author Response · Authors · 2024-06-07
>
> Thank you for your review, we respond to your points below.
>
> **W1.** The ability to aggregate features non-linearly through attentive pooling is beneficial for all methods, including previous image models such as Dinov2 and OpenCLIP and video models such as VideoMAE. We note that the benefits are more pronounced for video models such as VideoMAE and V-JEPA (see table 12 and table 13 in appendix). The advantage of attentive pooling is mostly an empirical observation, however, as pointed out by another reviewer, the use of attentive pooling is similar in spirit to the typical use of a [cls] token in previous works to extract a single embedding from transformer representations in both vision (e.g., ViT/DeIT, Dinov2) and language (BERT).
>
> **W2.**  As you mentioned, results on  SSv2, a task requiring fine-grained motion understanding, show that the video feature prediction objective leads to better motion representations than image models. In particular V-JEPA accuracy is 72.2% on SSv2 which is a 22.2 point improvement over I-JEPA and a 21.6 point improvement over DINOv2. This improvement is significant and is a strong signal demonstrating the superiority of video-pretrained models on motion-based tasks.  We agree that it would be interesting to explore other video tasks, in particular lower-level motion-based tasks, as well as forecasting and control tasks in future work.
>
> **W3.** Note that the ablations in Table 2 are intended to showcase the effect of the pretraining data distribution under a fixed computational budget. The table shows that the best downstream task performance is obtained by aligning the pretraining data distribution to the downstream task, but the model pre-trained on the VM2M distribution performs best on average. To further isolate the effect of data scale, we trained a ViT-L/16 and a ViT-H/16 on a 25% subset of VM2M for the same 90K optimization steps; the results are summarized below. Even with an identical computational budget, we see a positively monotonic trend on all tasks. However, it is almost surely the case that more training steps are required to better leverage larger data sources.
>
> *ViT-L/16 (90K optimization steps)*
> | | K400 | SSv2 | IN1k |
> | -- | -- | -- | -- |
> | **25%** | 72.7 | 66.1 | 73.9 |
> | **100%** | 73.7 | 66.2 | 74.8 |
> | | | |
>
> *ViT-H/16 (90K optimization steps)*
> | | K400 | SSv2 | IN1k |
> | -- | -- | -- | -- |
> | **25%** | 73.5 | 67.6 | 75.2 |
> | **100%** | 74.0 | 68.5 | 75.9
> | | | |
>
> **W4.** See Requested Change below.
>
> **RC1.** It is the latter option you have described: apply them separately to a duplicate sample and keep both in the same batch. We have clarified this in the appendix.
>
> **RC2.** Thank you for your comment. We added a citation to the paper “High fidelity visualization of what your self-supervised representation knows about on which the diffusion model is based.” (Bordes et al., 2021).
>
> **RC3.** We already perform an apples-to-apples comparison with reconstruction-based approaches in Table 1 where we control for the data distribution, model architecture, masking corruption, and training budget. In this table, pixel targets is the VideoMAE approach.
>
> **RC4.**  We compare V-JEPA with VideoMAEv2 in Table 6 using the attentive-pooling evaluation protocol.  As you indicate, we also compared using a finetuning evaluation protocol in Table 15 with various baselines, including VideoMAEv2.  We moved this table into the main body of the paper to provide better visibility to this comparison. Table 5 reports performance controlling for the architecture complexity using ViT-L or similarly sized architectures. We are not aware of a public/open ViT-L/16 model trained using VideoMAEv2.

---

> > ### Comment · Reviewer_qEpG · 2024-06-09
> > **Official Comment by Reviewer qEpG**
> >
> > Thank you for the explanations and editions.
> >
> > **RC1**: Thanks for clarifying it. Since it is the latter, is it correct to understand that when long-range masks are added to short-range masks, it will also double the batch size in training? Is the batch size the same when you compare different masking patterns? If so, could you please make it clear in the paper?
> >
> > **RC2**: Thanks for adding the citation, it will be more helpful if you can add the implementation details for that as well.
> >
> > **RC3**: Thanks. You mentioned that "pixel targets is the VideoMAE approach", while this remains unclear in the paper that if the EMA teacher/student update is kept when the pixel loss l is applied? If so, then it is not a strict VideoMAE? If not, it would still be good to add the comparison I suggested for fair comparison with Hiera.

---

> > > ### Author Response · Authors · 2024-06-11
> > >
> > > **RC1:** Yes, this is correct; while the batch size of the y-encoder is unchanged, the “batch size” of the x-encoder and predictor can be seen as doubled due to sampling two masks. All ablations in Table 4 are computed by sampling two masks for each video clip. We have clarified this in appendix B.
> > >
> > > **RC2:** We have added a more detailed description on how to apply RCDM for visualizing the representations in Appendix F.
> > >
> > > **RC3:** Yes, there is no EMA teacher when using the pixel targets in Table 1, thereby making it a “strict” VideoMAE baseline.
> > >
> > > Thank you again for all your detailed comments.

---

### Public Comment · ~Vimal_Thilak1 · 2024-11-05
**Updated Paper**

Hello AE,

Would it be possible to encourage the authors to consider uploading a deanonymized and updated version of their paper to TMLR? The attached PDF does not appear to adhere to camera ready guidelines.

Best

---

> ### Public Comment · ~Thomas_Palmeira_Ferraz1 · 2026-06-02
> **Update Paper**
>
> Hi, just wanted to raise here that the issue persists.

---

### Decision · Action_Editor_UNjr · 2024-07-01

**Recommendation:** Accept as is

**Comment:**

This is a good quality paper that can be a good reference for further work in representation learning from video. All reviewers suggest acceptance and are happy with the authors improvements post-review.

**Audience:**

There is considerable interest in learning visual representations from video and this will probably increase over time as progress accelerates.

**Claims And Evidence:**

All reviewers agree that results are well backed by experiments: "all of its claims are well supported by evidence", "scientifically robust", "good results supported by thorough experiments".